# Antibody toolkit reveals N-terminally ubiquitinated substrates of UBE2W

Christopher W. Davies [1,10], Simon E. Vidal[2,10], Lilian Phu[3,10], Jawahar Sudhamsu [4], Trent B. Hinkle[3], Scott Chan Rosenberg[2], Frances-Rose Schumacher[3], Yi Jimmy Zeng[3], Carsten Schwerdtfeger[5], Andrew S. Peterson[6], Jennie R. Lill[3], Christopher M. Rose[3], Andrey S. Shaw[7], Ingrid E. Wertz [2,8✉], Donald S. Kirkpatrick[3,9✉] & James T. Koerber [1✉]

The ubiquitin conjugating enzyme UBE2W catalyzes non-canonical ubiquitination on the N-termini of proteins, although its substrate repertoire remains unclear. To identify endogenous N-terminally-ubiquitinated substrates, we discover four monoclonal antibodies that selectively recognize tryptic peptides with an N-terminal diglycine remnant, corresponding to sites of N-terminal ubiquitination. Importantly, these antibodies do not recognize isopeptide-linked diglycine (ubiquitin) modifications on lysine. We solve the structure of one such antibody bound to a Gly-Gly-Met peptide to reveal the molecular basis for its selective recognition. We use these antibodies in conjunction with mass spectrometry proteomics to map N-terminal ubiquitination sites on endogenous substrates of UBE2W. These substrates include UCHL1 and UCHL5, where N-terminal ubiquitination distinctly alters deubiquitinase (DUB) activity. This work describes an antibody toolkit for enrichment and global profiling of endogenous N-terminal ubiquitination sites, while revealing functionally relevant substrates of UBE2W.

---

[1] Department of Antibody Engineering, Genentech, Inc., South San Francisco, CA, USA. [2] Departments of Molecular Oncology and Early Discovery Biochemistry, Genentech, Inc., South San Francisco, CA, USA. [3] Department of Microchemistry, Proteomics, and Lipidomics, Genentech, Inc., South San Francisco, CA, USA. [4] Department of Structural Biology, Genentech, Inc., South San Francisco, CA, USA. [5] Boston Biochem, a Bio-Techne Brand 840 Memorial Drive, Cambridge, MA, USA. [6] Department of Molecular Biology, Genentech, Inc., South San Francisco, CA, USA. [7] Research Biology, Genentech, Inc., South San Francisco, CA, USA. [8] Present address: Bristol Myers Squibb, 1000 Sierra Point Parkway, Brisbane, CA, USA. [9] Present address: Interline Therapeutics, South San Francisco, CA, USA. [10] These authors contributed equally: Christopher W. Davies, Simon E. Vidal, Lilian Phu. ✉email: ingrid.wertz@bms.com; dkirkpatrick@interlinetx.com; koerberj@gene.com

Protein ubiquitination is a complex post-translational modification that regulates diverse cellular functions including protein homeostasis, DNA damage response, innate and adaptive immunity, the cell cycle, and inflammatory signaling[1–4]. The covalent attachment of ubiquitin (Ub) to protein substrates occurs through the concerted activity of three enzymes: an E1 Ub-activating enzyme, an E2 Ub-conjugating enzyme, and an E3 Ub ligase[5–7]. Ub itself has seven lysine residues (K6, K11, K27, K29, K33, K48, and K63) and an N-terminus, all of which are amenable for conjugation[1]. K48 and K63-linked polyubiquitin chains are the most well-studied, with the traditional view being that K48-linked Ub chains mark proteins for proteasomal degradation, whereas K63-linked Ub chains have a protein scaffolding role[3,8,9]. Furthermore, studies show that mixed-linkage and branched chains exist[10,11] and can serve as more potent functional signals than homeotypic K48- or K63-linked Ub chains[11,12]. Conjugation of Ub to the ε-amino group of lysine residues is the most common form of ubiquitination. Other acceptor residues such as Thr, Ser, Cys, and the α-amino group of substrate N-termini have been identified and represent non-canonical ubiquitination targets[13–15]. However, the biological significance of these modifications is not well understood.

Upon its initial discovery, N-terminal Ub was posited to serve as a protein degradation signal[16–18]. These studies showed that engineered proteins lacking lysine residues or naturally occurring lysine-less proteins were still subject to proteasomal degradation, indirectly implicating N-terminal Ub as the degradation signal. Subsequent work demonstrated that N-terminally ubiquitinated proteins do not accumulate significantly upon proteasome inhibition, suggesting that N-terminal ubiquitination might have additional roles beyond promoting proteasome-mediated degradation[19], such as serving as a chaperone in the folding of nascent polypeptides[20]. With the exception of linear polyubiquitin chains formed by LUBAC, UBE2W is the only E2 Ub-conjugating or E3 ligase enzyme reported to form a peptide bond between the C-terminal Gly76 of Ub and the α-amino group of substrate protein N-termini[21,22]. In coordination with a ubiquitin ligase, current data suggest that UBE2W strictly mono-ubiquitinates protein substrates at their N-termini. These priming modifications can be elaborated by other E2/E3 complexes into N-terminally linked polyubiquitin chains[23]. Interestingly, UBE2W contains a partially disordered C-terminus that is critical for the recognition of substrates that have intrinsically disordered N-termini[24]. Despite our growing understanding of N-terminal ubiquitination and the structural and biochemical properties of UBE2W, only a small set of N-terminally ubiquitinated UBE2W substrates have been identified. New strategies to globally profile N-terminally ubiquitinated proteins are needed to further elucidate the physiological consequences of this modification.

Mass spectrometry (MS) is a powerful analytical tool for characterizing substrate-specific ubiquitination at the amino-acid residue level[25–27]. One key to success has been the generation of tools that specifically enrich peptides bearing Ub C-terminal remnants generated upon enzymatic cleavage. For example, the development of a monoclonal antibody (mAb) recognizing the tryptic Ub remnant consisting of isopeptide-linked diglycine attached to the sidechain of lysine (K-ε-GG) revolutionized the global profiling of ubiquitination sites[26,28,29]. More recently, a mAb was generated to recognize the extended LysC-generated remnant of Ub and distinguish between substrates conjugated to Ub from other Ub-like proteins such as NEDD8 and ISG15[19]. Other affinity-based enrichment or genetic tagging systems have also been developed[19,25,30–34]. Notably, a few of these strategies detect not only canonical K-ε-GG peptides but also peptides corresponding to N-terminal ubiquitination[19,30]. Previous quantitative proteomics data suggest that the relative abundance of

N-terminal Ub linkages is exceedingly low under basal conditions, given the frequency of lysines within a typical protein and the fact that ~80–90% of proteins can be acetylated on their N-termini, precluding N-terminal ubiquitination[19,35,36].

Here, we sought to develop an antibody toolset capable of specifically detecting and enriching the tryptic remnant unique to N-terminally ubiquitinated proteins as a complement to existing reagents and methods. We applied a rabbit immune phage strategy to discover four mAbs that selectively recognize peptides bearing an N-terminal diglycine-motif, but not the branched diglycine-remnant generated by trypsin digestion of ubiquitin conjugated lysines (K-ε-GG). Using a combination of biochemical and structural methods, we show that these mAbs predominately recognize the N-terminal diglycine with a relaxed selectivity for the third amino acid, enabling these mAbs to bind to a broad range of peptide sequences. We then established the capability of these mAbs to enrich modified peptides from digested cell lysates and designed experiments to elucidate N-terminal ubiquitination sites. As UBE2W is currently the only E2 known to mediate N-terminal ubiquitination of substrate proteins, we used an inducible *UBE2W* overexpression system and identified 73 putative UBE2W substrates, most of which are predicted to have disordered N-termini. Among these were two related deubiquitinases, UCHL1 and UCHL5, that were identified as targets of N-terminal ubiquitination; interestingly, we show that N-terminal ubiquitination is not a potent signal for degradation, but rather modulates deubiquitinase function. Overall, our studies provide previously undescribed insights into the enzymes, substrates, and biochemical effects of N-terminal ubiquitination.

## Results

**Generation of anti-GGX mAbs.** In order to better understand the role of N-terminal ubiquitination, we set out to discover antibodies capable of selectively enriching for tryptic peptides containing a diglycine sequence at their N-termini. Critically, we required that these mAbs possess minimal cross-reactivity to the conventional and more abundant K-ε-GG peptides, even though they share an identical diglycine sequence feature (Fig. 1a). We hypothesized that a sizeable pool of potential substrates would be nascent polypeptides with non-acetylated, intact initiator methionines and that upon trypsin digestion would yield peptides with a diglycine modification prior to methionine[37]. We, therefore, used a Gly-Gly-Met (GGM) peptide as an antigen for rabbit immunizations, as rabbits are known to generate high-affinity antibodies to peptides and small haptens[38]. Enzyme-linked immunosorbent assays (ELISAs) with purified polyclonal antibodies (pAb) confirmed a robust immune response against the GGM peptide, with surprisingly minimal cross-reactivity to peptides bearing K-ε-GG (Supplementary Fig. 1a).

Based on the strong pAb response, we proceeded to monoclonal antibody (mAb) discovery using phage display to directly select mAbs with the desired specificity. We constructed several single-chain Fv (scFv)-display libraries from individual rabbits and performed three rounds of plate-based biopanning against the GGM peptide with counterselection against the K-ε-GG peptide (Fig. 1b). After primary screening by phage ELISA, we sequenced hits and reformatted the unique clones into IgGs. In total, we identified four unique antibody clones (1C7, 2B12, 2E9, and 2H2) with high sequence similarity, but with diversity occurring in multiple complementarity-determining regions (CDRs) (Supplementary Fig. 1b). We characterized the mAbs by ELISA against the GGM and K-ε-GG peptides and found that all four clones selectively bound to GGM and not K-ε-GG peptides (Fig. 1c).

Although the largest pool of potential N-terminally ubiquitinated mAb targets is comprised of nascent polypeptides that

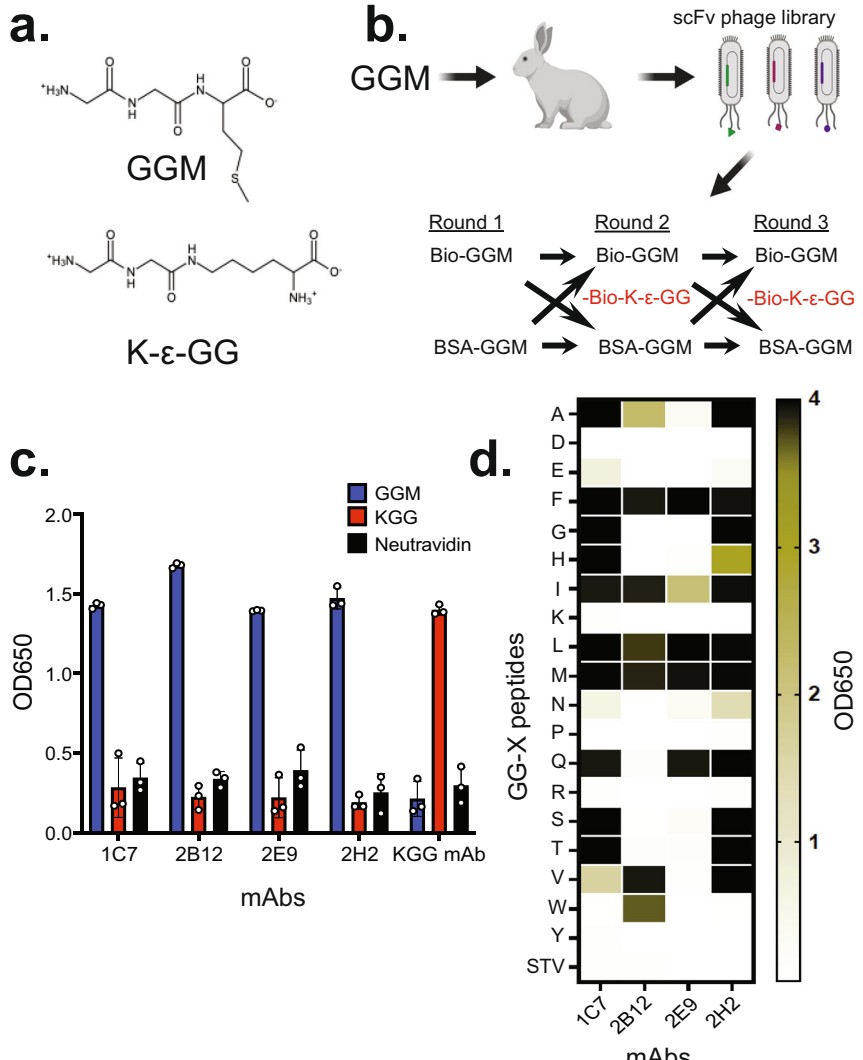

**Fig. 1 Discovery and characterization of anti-GGX mAbs. a** The chemical structure of GGM and K-ε-GG peptides highlights the high similarity between both peptides. **b** Schematic of the immunization and phage panning strategy used to discover the anti-GGX mAbs. Biotinylated (Bio-GGM) and BSA conjugated (BSA-GGM) were used for positive selections and a biotinylated K-ε-GG (red, Bio K-ε-GG) was used for negative selections. **c** ELISA characterization of the four anti-GGX mAbs and anti-K-ε-GG mAb confirms the GGM specificity of the antibodies. Binding to the GGM peptide is shown in blue, K-ε-GG peptide in red, and neutravidin in black. Results are representative of biological replicates ($n = 3$). Data are presented as mean ± SD. **d** ELISA characterization of anti-GGX mAbs against GGX peptides, in which X represents all twenty amino acids except for cysteine, shows degenerate recognition at the X position. The darker shading corresponds to the better binding. STV is the streptavidin control. Results are representative of biological replicates ($n = 3$). Data are presented as mean ± SD. Source data are provided as a Source Data file.

predominantly start with methionine in eukaryotes, several other sources of free N-termini are present. These sources result from Met-clipping via aminopeptidases, signal peptide removal, and internal proteolysis. In the case of clipping by Met aminopeptidases (MetAP), cleavage typically occurs before Ala, Cys, Gly, Pro, Ser, Thr, or Val residues[39]. To probe whether our mAbs would also recognize tryptic peptides from these potential sites of N-terminal ubiquitination, we evaluated peptides containing diglycine followed by each of the 20 amino acids except cysteine. We denote these as GGX peptides, where X represents the initial amino acid in a polypeptide sequence that contains a GG sequence addition extending from the N-terminus. Notably, mAbs 1C7 and 2H2 recognize a similar set of GGX peptides, whereas 2E9 and 2B12 exhibit different specificities. Collectively, these four mAbs bind 14 of the 19 GGX peptides, showing a strong preference for several amino acids that would be susceptible to MetAP clipping (Gly, Ala, Ser, Thr, and Val) (Fig. 1d)[39]. Overall, these data reveal

that we have discovered four anti-GGX mAbs that selectively recognize tryptic diglycine-containing linear peptides with broad specificity at the third position (X) and no cross-reactivity to isopeptide-linked diglycine-modified lysine containing peptides that correspond to canonical ubiquitination.

**Structural basis for GGX peptide recognition.** To gain insight into the exquisite selectivity for linear diglycine-containing peptides, we determined the x-ray crystal structure of the 1C7 Fab bound to a GGM peptide at 2.85 Å resolution (Supplementary Table 1). There are two Fab-GGM complexes in the asymmetric unit, with well-defined electron density for the GGM peptide that binds in a pocket at the interface of the heavy chain (HC) and light chain (LC) CDRs (Supplementary Fig. 2a, b). The interaction of a GGM peptide and Fab has a buried surface area of 247.5 Å² (Pymol). Interestingly, this pocket at the LC-HC interface is

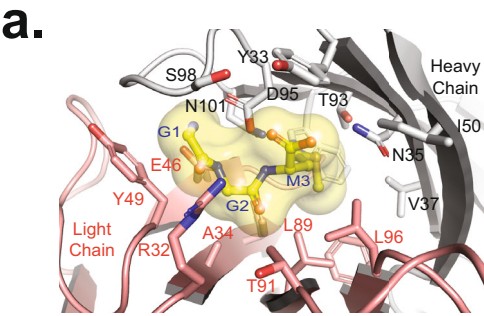

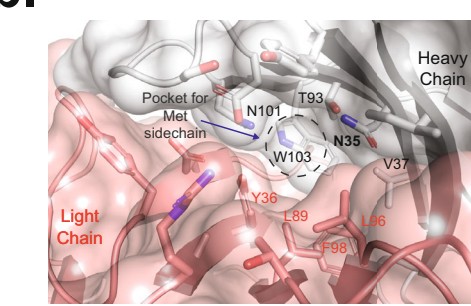

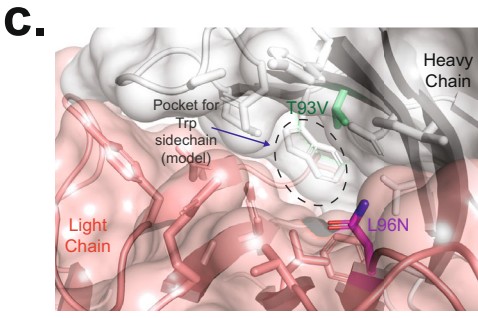

**Fig. 2 Structural analysis of anti-GGX Fab bound to GGM peptide. a** A detailed view of the interaction between the GGM peptide and the 1C7 Fab shows that the peptide resides within a pocket at the interface between the Fab heavy and light chain. GGM peptide is shown in yellow sticks with a transparent surface. The HC residues are shown as white sticks, and LC residues are shown as pink sticks. **b** A detailed view of the methionine sidechain recognition pocket located at the LC-HC interface, containing a mixture of hydrophobic and hydrophilic residues. The GGM peptide has been removed for clarity. The HC and LC residues are colored as before. The dashed circle highlights the pocket in which the GGM Met sidechain resides. **c** A structural model of the GGW Trp peptide pocket reveals two residue differences between 1C7 and 2B12 that may lead to the selective recognition of GGW by 2B12. The GGW peptide has been removed for clarity. The HC and LC residues are colored as before. The HC Thr93Val substitution is shown in green sticks, and the LC Leu96Asn substitution is shown in magenta sticks. The dashed circle highlights the pocket in which the GGW Trp sidechain would reside.

commonly used by antibodies to recognize haptens[40]. Close inspection of the Fab-peptide complex reveals a series of hydrogen bonds that facilitate recognition of the diglycine portion of the peptide. The sidechains of HC Asp95 and LC Glu46 make five hydrogen bonds with the backbone of the diglycine, including with the amino terminus and the two amides (Fig. 2a). The negative charge of the two carboxylates neutralizes the positive charge of the amino terminus, which is surrounded by the Fab residues and excluded from the solvent. In addition, tight packing of diglycine against LC Arg32, LC Ala34, and LC Tyr49 likely

sterically blocks recognition of a non-Gly residue at either of the first two positions in the peptide. This binding mode is further stabilized by a hydrogen bond between the HC Asp95 sidechain and backbone amine of Met (Fig. 2a).

Inspection of the Met-binding pocket reveals the structural basis for the degenerate amino-acid specificity at this position. This pocket is lined with HC residues Asn35, Val37, Thr93, Asn101, and Trp103 on one side, and LC residues Tyr36, Leu89, Leu96, and Phe98 on the other side (Fig. 2b). The closest contact made with the methionine sidechain is a 3.1 Å hydrogen bond between the sulfur atom and HC Thr93 (Fig. 2a). This analysis indicates a loosely packed pocket with both hydrophobic and hydrophilic character, which enables the recognition of a broad set of amino acids. Lack of recognition of Trp, Lys, Tyr, and Arg is readily explained by steric clashes with multiple sidechains that line this pocket. In the case of Pro, multiple clashes between the HC Asp95 and LC Tyr35 sidechains in the antibody and the Pro sidechain in the peptide would occur (Supplementary Fig. 2c). A key feature of this antibody is the lack of recognition of the highly similar K-ε-GG peptides. We, therefore, analyzed how a K-ε-GG peptide would interact with the Fab assuming the same mode of GG recognition as for the GGM peptide. We hypothesized that the lysine sidechain could follow a similar trajectory as the main chain of GGM, however, branching at the Lys Cα position (i.e., residues before and after Lys in the peptide) would sterically clash with CDRH3 and HC Tyr33 preventing binding to the mAb.

As both 2E9 and 2B12 mAbs have sequence similarity to 1C7, but exhibit altered recognition profiles, we analyzed the potential structural basis for this result. Within the GGM-binding pocket, 2E9 has two differences (LC Thr91Glu and Leu96Phe), which would compact the Met pocket, preventing recognition of a broad set of residues (Fig. 2a). Out of all the residues that differ between 1C7 and 2B12, we identified four residues in the LC (A34, Y49, T91, and L96), and three residues in the HC (Y32, I50, and T93) that directly interact with the GGM peptide. When we model 2B12 binding to GGW based on the 1C7 structure, it appears that only LC Leu96Asn and HC Thr93Val play significant roles in Trp recognition in a GGW peptide. LC Leu96Asn appears to slightly increase the volume and hydrophilicity of the pocket, whereas, HC Thr93Val increases the hydrophobicity of the pocket, without significantly altering the volume (Fig. 2c). Collectively, our structural studies elucidate how these antibodies achieve degenerate recognition of GGX while avoiding recognition of the highly similar K-ε-GG.

**Anti-GGX mAbs selectively enrich GGX peptides from cell lysates.** With our biochemical and structural data confirming in vitro selectivity of the mAbs for linear tryptic GGX peptides, we next investigated their capability for enriching peptides from complex cell lysates. Lysates from unstimulated HEK293 cells were digested with trypsin and immunoaffinity enrichment was performed for GGX peptides using each of the four anti-GGX mAbs individually. The resulting peptide pools were analyzed by liquid chromatography-tandem mass spectrometry (LC-MS/MS) (Fig. 3a). In parallel, immunoaffinity enrichment was performed with the anti-K-ε-GG mAb as a control. Given the high abundance of K48- and K63-linked Ub chains present in lysates, we used those target peptides to confirm the selectivity of our mAbs for immunoaffinity enrichment of GGX peptides overabundant K-ε-GG peptides. Extracted ion chromatograms (XIC) for representative peptide ions corresponding to K48- and K63-linked Ub chains were prepared from LC-MS data for the anti-GGX and anti-K-ε-GG-enriched samples to compare their levels. Compared with the anti-K-ε-GG mAb, which showed strong enrichment of isopeptide-linked K48 and K63 Ub peptides, no

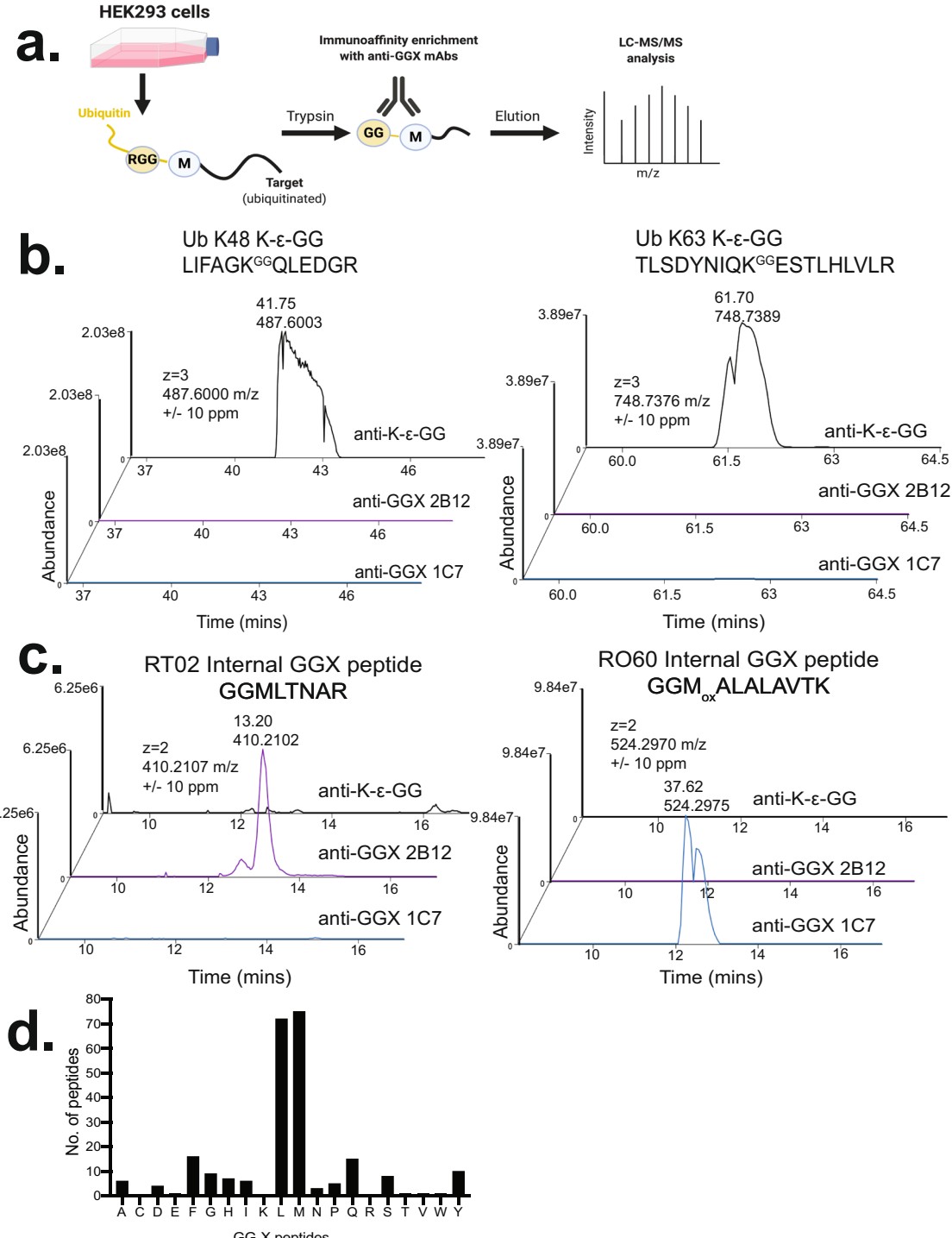

**Fig. 3 Anti-GGX mAbs selectively IP GGX peptides and not K-ε-GG peptides. a** Experimental schema depicting lysate preparation, trypsin proteolysis, immunoaffinity enrichment of GGX peptides followed by quantitative mass spectrometry (LC-MS/MS) analysis. Portions of ubiquitin are shown in yellow. **b** Extracted ion chromatograms (±10 ppm) for K48 and K63 K-ε-GG polyubiquitin chain linkage peptides LIFAGK$^{GG}$QLEDGR and TLSDYNIQK$^{GG}$ESTLHLVLR in anti-K-ε-GG, anti-GGX 2B12, and anti-GGX 1C7 immunoaffinity enrichment MS experiments. **c** Extracted ion chromatograms (±10 ppm) for internal GGX peptides GGMLTNAR and GGM$_{ox}$ALALAVTK in anti-K-ε-GG, anti-GGX 2B12, and anti-GGX 1C7 immunoaffinity enrichment MS experiments. **d** Analysis of the X position of immunoaffinity enriched internal GGX peptides indicates a preferential enrichment for GGM and GGL peptides with other amino acids observed at lower frequencies.

signal was detected when any of the four anti-GGX mAbs was used for enrichment (Fig. 3b and Supplementary Fig. 3a).

Next, we investigated peptide sequences that were enriched by the anti-GGX mAbs. Given the frequency of glycine, lysine, and arginine residues, many GGX peptides are encoded in the

proteome, stemming from proteins that contain naturally occurring internal GGX sequence motifs preceded by a trypsin cleavage site (R/KGGXXXX). As predicted, many such peptides were detected in this experiment. Notably, XICs for representative internal GGX sequences applied across each enriched sample

demonstrated specific signal in anti-GGX mAb-enriched samples, but none following immunoaffinity enrichment with the anti-K-ε-GG mAb. Combined with the ELISA data, these results confirm the selectivity of the anti-GGX mAbs for the sequences being targeted (Fig. 3c and Supplementary Fig. 3b).

Internal GGX peptides, like N-terminal ubiquitination sites, only become exposed after trypsin cleavage and provide valuable insights into the sequence preferences of each mAb. Using these internal peptides, we profiled the amino-acid preference at the third position. Consistent with the panning strategy, ELISA, and structural data, we observe a strong preference for methionine and leucine at the third position, with the next most prevalent amino acids being phenylalanine and glutamine (Fig. 3d). To further profile sequence specificity, we also generated sequence logos for each anti-GGX mAb[41]. Again, we see a preference for methionine and leucine at the third position, but importantly, a diversity of other amino acids at positions 3–6 (Supplementary Fig. 3c). These data reconfirmed ELISA results showing that each mAb enriched for a unique set of peptides with partial overlap between individual mAbs, especially when considering the differences in positions 4–6 (Supplementary Table 2). Based on these data, we established a PTMscan® protocol using an equimolar mixture of the four anti-GGX mAbs for subsequent MS experiments to ensure the broadest peptide coverage.

Focusing on sites of N-terminal ubiquitination, we manually inspected the data and filtered peptide spectral matches (PSMs) for those bearing a diglycine remnant at the initiator methionine, or neo-N-terminus. We searched for peptides bearing a 114.0429 Da mass addition, corresponding to the mass of diglycine, and then, confirmed that the genome-encoded polypeptide sequence did not contain a diglycine sequence immediately preceded by a trypsin sensitive R/K residue. Following rigorous filtering, we identified eight putative N-terminal ubiquitination sites (Supplementary Data 1). One example was a putative N-terminal ubiquitination site observed on Serine/threonine-protein kinase 11-interacting protein (S11IP) (Fig. 4a, b), in addition to several sites that have been previously described[19]. Overall, this demonstrated the selective ability of these anti-GGX mAbs to enrich a broad panel of both internal, genome-encoded GGX peptide sequences exposed by trypsin digestion, and GGX peptides stemming from N-terminal ubiquitination.

**Proteomic identification of putative UBE2W substrates**. The pilot MS experiment validated the utility of the anti-GGX mAbs, but yielded only eight putative N-terminally ubiquitinated substrates from endogenous HEK293 cells. This result agreed with the low basal levels of this modification reported in the literature[19]. As UBE2W is the only E2 Ub-conjugating enzyme known to mediate N-terminal ubiquitination and UBE2W expression levels are low in HEK293 cells, we reasoned that exogenous expression of UBE2W might stimulate N-terminal ubiquitination of endogenous substrates. We, therefore, generated a doxycycline (Dox)-inducible UBE2W HEK293 cell line (Supplementary Fig. 4a) and performed a similar workflow as in the pilot MS experiment. Here we applied label-free quantification (LFQ) of MS1 peak intensities to compare +Dox UBE2W expression to control -Dox conditions with the aim of identifying UBE2W substrates as those proteins from which GGX peptides at their N-termini increased in abundance upon UBE2W expression. Applying a similar approach as in the pilot, we filtered PSMs for protein N-terminal sequences corresponding to diglycine attachment at either the initiator methionine or neo-N-terminus. In total, we found 152 unique GGX PSMs derived from 109 proteins at peptide and protein false discovery rates of 0.80% and 3.67%, respectively. Peptides stemming from TrEMBL database entries (non-curated sequences in UniProt) encoding 'X' as the

first amino acid was subsequently excluded from consideration. Using a criterion of $\log_2$-fold change ($\log_2$FC) > 1 and $p < 0.05$ for PSMs, this experiment reported 32 UBE2W substrates (Fig. 5a and Supplementary Data 1).

Most E2 Ub-conjugating enzymes work cooperatively with E3 ligases, and previous work reported that UBE2W exhibited RNF4-dependent ubiquitination of some substrates[23]. Therefore, we generated Dox-inducible RNF4 and bicistronic (RNF4/ UBE2W, "combo") expression vectors and prepared stable HEK293 cell lines (Supplementary Fig. 4b). We then performed an anti-GGX mAb immunoaffinity enrichment, this time in concert with isobaric multiplexing via tandem mass tagging (TMT) as has been described for anti-K-ε-GG mAbs[42], in addition to the LFQ approach taken above. In the TMT analysis, it was possible to compare several replicates of each condition against each other in a single multiplexed experiment: control (-Dox), UBE2W only, RNF4 only, and RNF4/UBE2W (combo). This set of samples made it possible to evaluate potential E2/ E3 synergy between UBE2W and RNF4 in the N-terminal ubiquitination of substrates. In this paradigm, UBE2W substrates are represented in two of the contrasts: UBE2W-Control and Combo-RNF4. Contrast refers to a pair of conditions being compared across the list of identified and quantified features (Supplementary Data 1). In the TMT analysis, we identified 141 unique N-terminal GGX PSMs derived from 99 proteins at peptide and protein false discovery rates of 0.80% and 2.02%, respectively. Examination of the data revealed that RNF4 overexpression did not significantly affect N-terminal ubiquitination levels either in the RNF4 only samples or synergistically when co-expressed with UBE2W (i.e., in the combo samples). Each of these conditions yielded similar quantitative data as the control and UBE2W only conditions, respectively. To look for hits emerging from multiple conditions, we compared the $\log_2$FC of the Combo-RNF4 contrast to that of UBE2W-Control, yielding a high confidence set of 60 UBE2W substrates with $\log_2$FC > 1 and $p < 0.05$ across multiple conditions. (Fig. 5b and Supplementary Data 1).

In the corresponding LFQ analysis, 186 unique N-terminal GGX PSMs derived from 120 proteins were identified at a peptide false discovery rate of 1.38%. The protein false discovery rate of this data set was abnormally high at 13.33%, owing to the frequency of repeat identifications at the peptide level. Focusing on proteins whose N-terminal ubiquitination levels increased ($\log_2$FC > 1 and $p < 0.05$) in UBE2W only versus control, Combo versus RNF4 only, and Combo vs Control, this data yielded 38 UBE2W substrates that partially overlap with the TMT analysis (Supplementary Fig. 4c). Filtering for the highest confidence hits by requiring $\log_2$FC > 1 and $p < 0.05$ in both UBE2W-control and Combo-RNF4 only contrasts yielded 28 high confidence UBE2W substrate protein hits (Fig. 5c). Integrating all of our MS experimental data revealed significant overlap in the identified substrates, with a subset of unique substrates identified in each of the individual experiments (Fig. 5d). Further inspection revealed that the majority (~53%) were shared between the UBE2W alone and the UBE2W/ RNF4 Combo conditions, confirming that exogenous expression of RNF4 did not enhance the activity of UBE2W in cells (Supplementary Fig. 4c and Supplementary Data 1).

Collectively from three quantitative immunoaffinity enrichment experiments: LFQ interrogating UBE2W overexpression versus control, TMT analysis comparing UBE2W and RNF4 overexpression individually and in combination versus control, and the follow-up LFQ experiment, 73 UBE2W substrates are reported as reaching statistical significance (Supplementary Data 1). Based on the quantitative data arising from the TMT experiment, we saw consistently increased signals for several proteins from samples expressing UBE2W compared with the

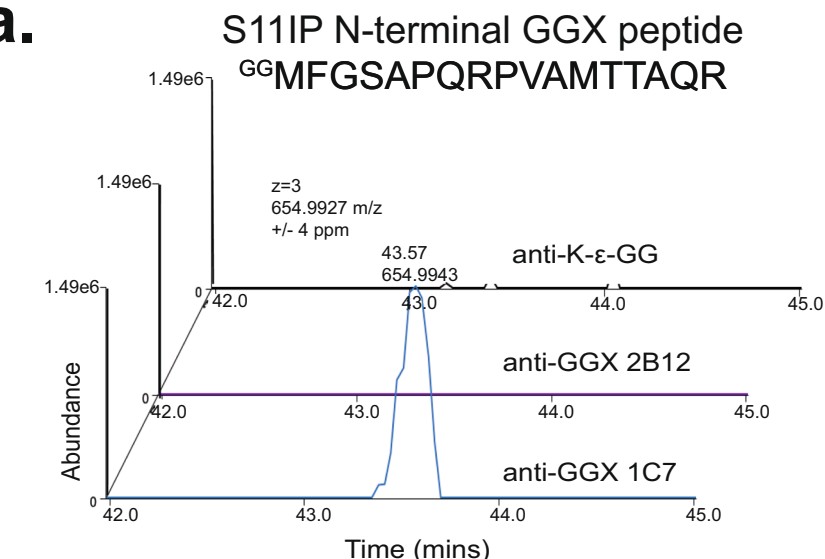

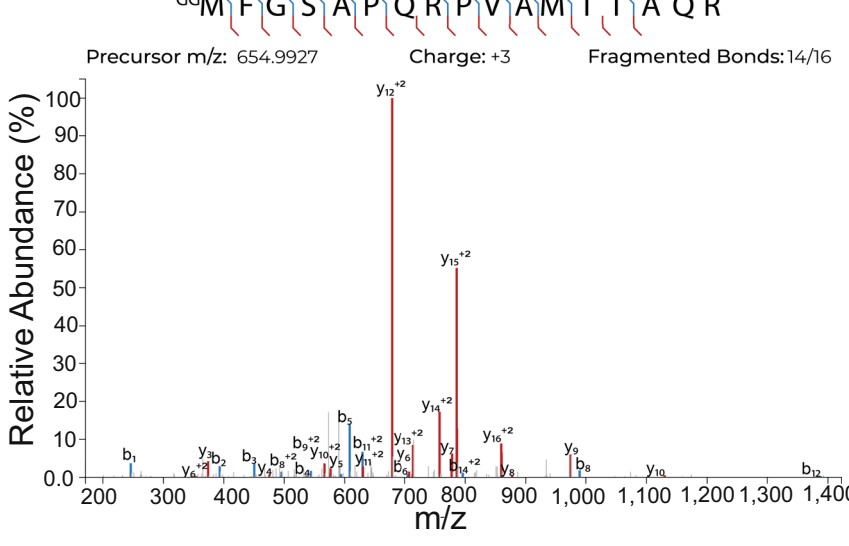

**Fig. 4 MS identification of S11IP N-terminal Ub peptide. a** Extracted ion chromatograms (±4 ppm) for N-terminal GGX peptide GGMFGSAPQRPVAMTTAQR in anti-K-ε-GG, anti-GGX 2B12, and anti-GGX 1C7 immunoaffinity enrichment MS experiments. **b** MS/MS spectrum identification of triply charged 654.9938 m/z N-terminal GGX modified peptide GGMFGSAPQRPVAMTTAQR. Detected b- and y- ions highlighted in blue and red, respectively.

controls, giving us confidence that these are indeed substrates of UBE2W (Supplementary Fig. 4d).

Next, we sought to validate some of the identified substrates by ectopically expressing lysine-less C-terminally HA-tagged proteins with *UBE2W* or *UBE2W^W144E*, a mutant that reduces ubiquitin binding, in cells[24]. Using C-terminally tagged substrates is critical given the previous observation that an N-terminal HA tag represents an intrinsically disordered sequence that can be recognized and modified by UBE2W[24]. Most of our identified UBE2W substrates showed enrichment of the monoubiquitinated form; however, a few substrates showed high molecular weight bands consistent with polyubiquitination that we posit occurs subsequent to N-terminal ubiquitination through the actions of other enzymes (Fig. 5e). Importantly, monoubiquitinated species

of these substrates did not accumulate in cells expressing the mutant *UBE2W^W144E*, confirming that these substrates depend upon UBE2W ubiquitin binding and subsequent transfer to the target protein amino terminus (Fig. 5e). Inspection of the identified substrates across experiments revealed that N-terminal ubiquitination occurred exclusively on the translation-initiating methionine (Supplementary Data 1). This was surprising because many of these same proteins display small, hydrophobic amino acids in the second position which tend to trigger the removal of the N-terminal Met by MetAP[39] (Supplementary Fig. 4e). As UBE2W is known to preferentially ubiquitinate proteins with disordered N-termini[24], we used protein prediction software to assess the frequency of disordered N-termini amongst these putative substrates. Using Protein DisOrder prediction System[43],

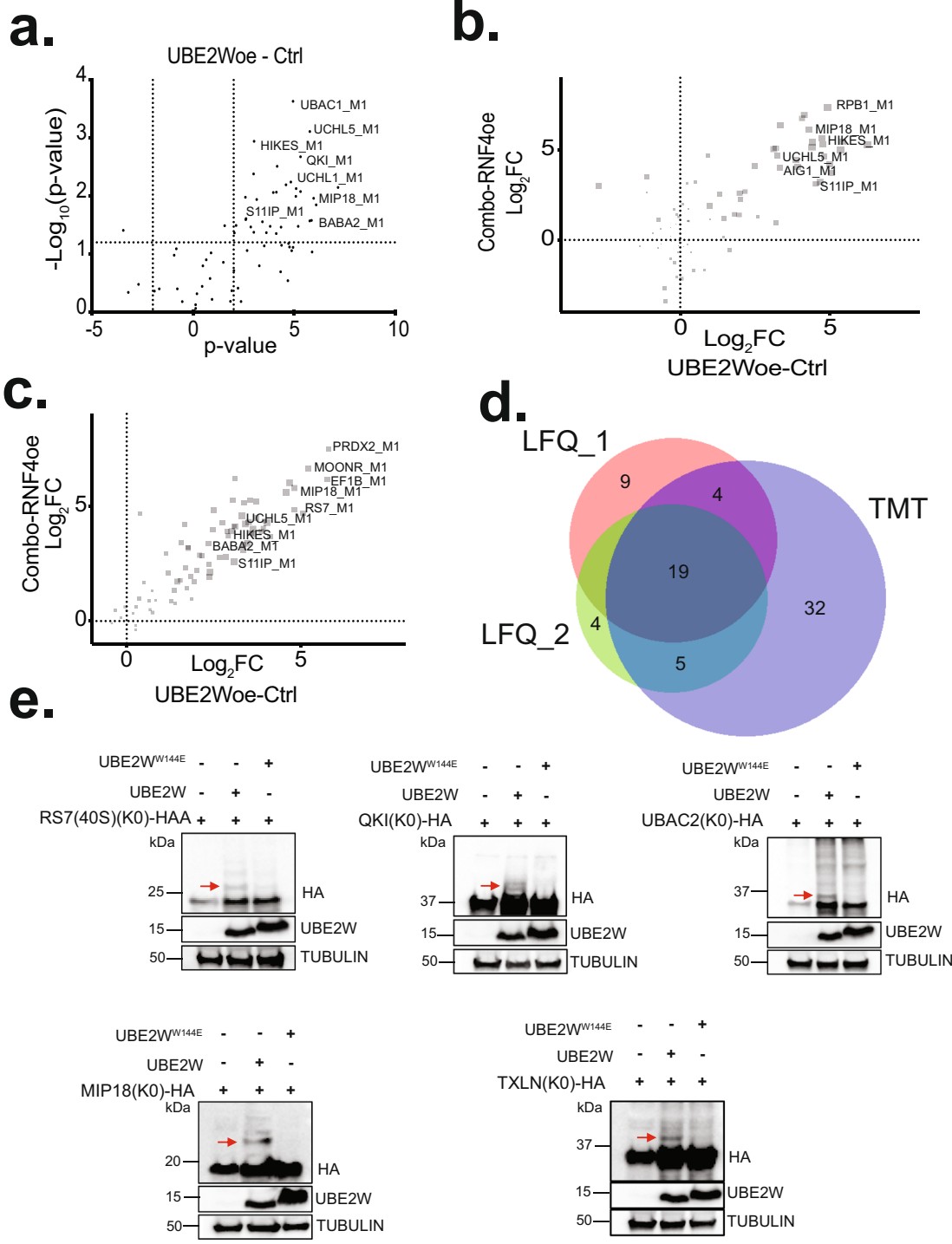

we found that 61 (~84%) of these proteins have predicted disordered N-termini (Supplementary Data 1). Taken together, we have identified 73 cellular substrates of UBE2W.

**UCHL1 and UCHL5 are substrates of UBE2W**. Notable among the UBE2W substrate list were two members of the ubiquitin C-terminal hydrolase (UCH) family of DUBs, UCHL1, and UCHL5 (Fig. 6a, b, Supplementary Fig. 5a, b, and Supplementary Data 1). UCHL1 was identified as a putative substrate in one of three LC-MS experiments, whereas UCHL5 was identified in all

three. Owing to the idiosyncratic nature of data-dependent shotgun sequencing, data to demonstrate N-terminal ubiquitination of UCHL1 was unfortunately absent from the TMT analysis (See Methods section). Two distinct forms of N-terminally ubiquitinated UCHL1 and UCHL5 peptides were identified, representing the semi-tryptic (Supplementary Fig. 5a, b) and fully tryptic (Fig. 6a, b) forms of each, further increasing our confidence in their respective identifications. Previously, it had been suggested that UCHL1 is N-terminally ubiquitinated, however the enzymes responsible for this modification have not been characterized[44]. To validate that these two DUBs were indeed N-

**Fig. 5 Identification of N-terminal ubiquitination sites mediated by UBE2W. a** Volcano plot showing differential N-terminal protein ubiquitination data for UBE2Woe versus control conditions in label-free GGX-MS experiment. Each data point represents an N-terminally modified GGX modified protein with a representative set of proteins shown (UniProt human). Protein-level cutoffs set at $\log_2$ fold change (FC) > 1.0 and $-\log_{10} P$ value > 1.3 ($P < 0.05$) are marked by dashed lines. **b** Scatter plot showing proteins with differential N-terminal protein ubiquitination in "UBE2Woe versus Control" and "Combo versus RNF4oe" contrasts in TMT-11-plex GGX-IAP-LC-MS/MS experiment. Sizes of the data points are scaled with $P$ values. All experiments were performed with replicates (control $n = 3$, UBE2W only $n = 3$, RNF4 only $n = 2$, and combo $n = 3$). **c** Scatter plot showing proteins with differential N-terminal protein ubiquitination in "UBE2Woe versus Control" and "Combo versus RNF4oe" contrasts in label-free GGX-MS experiment. Sizes of the data points are scaled with $P$ values. For **a–c**, R package MSstats was used for statistical analysis to perform differential abundance analysis. MSstats estimated log2(fold change) and the standard error by linear mixed effect model for each protein. To test the two-sided null hypothesis of no changes in abundance, the model-based test statistics were compared to the Student $t$ test distribution with the degrees of freedom appropriate for each protein and each data set. **d** An area proportional Venn diagram comparing the number of identified putative UBE2W substrates from each of the three MS experiments. The first LFQ experiment is the red circle, the second LFQ experiment is the green circle, and the TMT experiment is the purple circle. The numbers represent the number of substrates that were identified within each experiment or shared between multiple experiments. The figure was generated from the BioVenn web application[73]. **e** Western blots of WT, doxycycline-inducible *UBE2W/RNF4*, and doxycycline-inducible *UBE2W^{W144E}/RNF4* stable HEK293 cells transfected with constructs encoding for five lysine-less mutants of putative UBE2W substrates. An HA tag was fused at the C-terminus of each construct for protein detection using an anti-HA tag antibody. The red arrows indicate the modified forms of each substrate. Results are representative of three independent experiments. Source data are provided as a Source Data file.

terminally ubiquitinated, we established in vitro ubiquitination assays with purified proteins and observed that UBE2W can monoubiquitinate lysine-less versions of both UCHL1 and UCHL5 (Fig. 7a). Supporting these data, we saw that endogenous UCHL1 was monoubiquitinated upon expression of *UBE2W* in cells (Fig. 7b). Importantly, *UBE2W^{W144E}* expression did not support the formation of Ub-UCHL1 (Fig. 7b). We were unable to see the modification of endogenous UCHL5 in western blot experiments. To confirm that we had identified these substrates by directly binding the GG-modified peptide, and not the unmodified N-terminal peptides of UCHL1 and UCHL5, we generated N-terminal tryptic peptides that correspond to the N-terminus of UCHL1 and UCHL5 to confirm mAb binding by ELISA. We included peptides corresponding to linear ubiquitin after trypsin digestion as an additional positive control. As expected, we saw that the anti-GGX mAbs only bind to the peptides that are GG-modified on the initiator Met, and not the unmodified N-terminal sequence (Supplementary Fig. 5c). In addition, we show that the anti-GGX mAbs can enrich the proteotypic peptide representative of linear ubiquitin chains, although the conditions in our cellular experiments were not suitable for generating linear ubiquitin chains.

N-terminal ubiquitination has been proposed to be a signal for protein degradation[15–18]. However, it was recently shown that N-terminally ubiquitinated proteins accumulate only modestly in the presence of proteasome inhibitors, suggesting that the primary role for N-terminal ubiquitination may not be to facilitate protein degradation[19]. Therefore, we evaluated whether N-terminal ubiquitination promotes the degradation of UCHL1 in cellular assays. To test this, we expressed *UBE2W* and treated HEK293 cells with the proteasome inhibitor Bortezomib (Btz). Although high molecular weight ubiquitinated proteins accumulated in cells treated with Btz, we did not observe an accumulation of Ub-UCHL1 (Supplementary Fig. 5d). To further validate that the N-terminal monoubiquitination of UCHL1 does not trigger its degradation, we performed cycloheximide chase experiments. Although the labile protein p21 was quickly degraded in cells expressing *UBE2W* and treated with cycloheximide, Ub-UCHL1 remained stable (Supplementary Fig. 5e). UCHL1 protein levels only started to decline at a later time point (5 h) (Supplementary Fig. 5e). Altogether, these results suggest that N-terminal monoubiquitination by UBE2W in cells does not trigger the degradation of UCHL1.

**N-terminal ubiquitination regulates UCHL1 and UCHL5 DUB activity.** As N-terminal ubiquitination did not appear to promote

UCHL1 degradation in our cell-based assays, we next evaluated whether N-terminal ubiquitination modulated UCHL1 and UCHL5 deubiquitinase function. To test this idea, we generated a number of UCHL1 and UCHL5 variants including wild-type (UCHL1^{WT} and UCHL5^{WT}), catalytically inactive mutants (UCHL1^{C90S} and UCHL5^{C88S}), N-terminal Ub mimetics (Ub^{G76V}-UCHL1 and Ub^{G76V}-UCHL5), and N-terminal Ub mimetics that are no longer capable of binding to ubiquitin-binding domains (UBDs) (Ub^{I44A,G76V}-UCHL1 and Ub^{I44A,G76V}-UCHL5)[45]. For the mimetics, we fused the C-terminus of ubiquitin to the initial methionine of the DUB, while mutating the Gly76 in ubiquitin to valine to prevent removal of ubiquitin via deubiquitinase activity.

Previous structural work showed that Ub-binding causes UCHL1 active site residues to rearrange into a catalytically competent configuration[46]. However, monoubiquitination of internal lysines near the active site of UCHL1 has been shown to block the binding of its substrates[44]. Moreover, the activity of UCHL5 is tuned at the level of substrate affinity[47]. Therefore, we wondered if N-terminal ubiquitination would modulate the ubiquitin-binding affinities of UCHL1 and UCHL5. Using bio-layer interferometry (BLI), we investigated the monoubiquitin binding capabilities of UCHL1^{WT}, UCHL5^{WT}, Ub^{G76V}-UCHL1, Ub^{G76V}-UCHL5, Ub^{I44A,G76V}-UCHL1, and Ub^{I44A,G76V}-UCHL5 (Fig. 8a). Consistent with previous reports[48,49], we observed a strong interaction between monoubiquitin and UCHL1^{WT}. However, binding was only observed between monoubiquitin and Ub^{G76V}-UCHL1 at high monoubiquitin concentrations (5 μM) (Fig. 8b and Supplementary Fig. 6a). Interestingly, Ub^{I44A,G76V}-UCHL1 showed a approximately threefold increase in binding compared with Ub^{G76V}-UCHL1, suggesting that UCHL1 is able to interact in cis with its N-terminal Ub modification. (Fig. 8b and Supplementary Fig. 6a). Similar trends were seen for UCHL5, however, the affinity for monoubiquitin was much reduced compared to UCHL1, and I44A showed no effect (Fig. 8b and Supplementary Fig. 6b). Thus, we conclude that N-terminal ubiquitination prevents UCHL1 and UCHL5 from binding monoubiquitin.

Next, we asked whether the deubiquitinase activities of UCHL1 and UCHL5 were altered upon N-terminal ubiquitination by performing DUB activity assays using ubiquitin-rhodamine 110 (Ub-Rho110). The kinetics of UCHL1^{WT} and UCHL5^{WT} agreed with previous reports[46,47], and as expected, we saw no activity from catalytically dead UCHL1^{C90S} and UCHL5^{C88S} (Fig. 8c and Table 1). Strikingly, the N-terminal ubiquitination of UCHL1 and UCHL5 conferred opposite effects on their respective

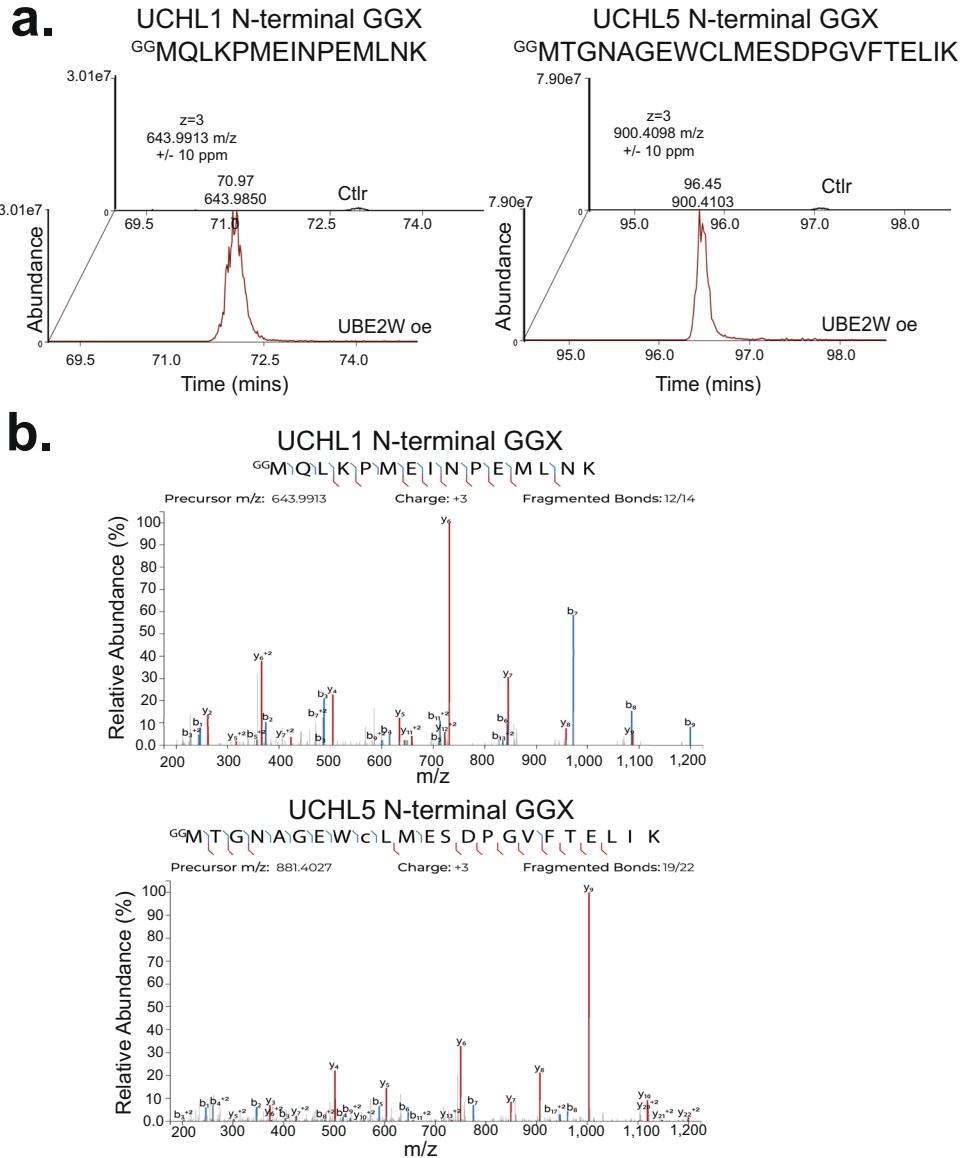

**Fig. 6 MS identification of UCHL1 and UCHL5 N-terminal Ub peptides. a** Extracted ion chromatograms (±10 ppm) for N-terminal tryptic GGX peptides $^{GG}$MQLKPMEINPEMLNK and $^{GG}$MTGNAGEWCLMESDPGVFTELIK of UCHL1 and UCHL5, respectively, in control and UBE2Woe conditions from GGX-IAP-LC-MS/MS experiment. **b** MS/MS peptide spectral matches for N-terminal tryptic GGX modified peptides $^{GG}$MQLKPMEINPEMLNK (triply charged, 643.9907 m/z) and $^{GG}$MTGNAGEWCLMESDPGVFTELIK (triply charged, 900.4094 m/z). Detected b- and y- ions highlighted in blue and red, respectively.

deubiquitinase activities. Ub$^{G76V}$-UCHL1 showed significantly reduced activity compared to UCHL1$^{WT}$, whereas Ub$^{G76V}$-UCHL5 had a significantly enhanced activity compared with UCHL5$^{WT}$ (Fig. 8c and Table 1). Interestingly, the I44A mutation in ubiquitin showed minimal difference in UCHL1 and UCHL5 activity (Fig. 8c and Table 1). To corroborate the Ub-Rho110 assays, we utilized the suicide probe ubiquitin-vinyl sulfone (Ub-VS)[50]. We observed that UCHL1$^{WT}$ readily reacts with Ub-VS, nearing completion after 30 min, whereas, Ub$^{G76V}$-UCHL1 remained largely unmodified, whereas, Ub$^{I44A,G76V}$-UCHL1 shows a slight increase in activity compared to the wild-type Ub, but far less than UCHL1 alone (Fig. 8d). Conversely, UCHL5$^{WT}$ was only partially modified at 30 min, whereas Ub$^{G76V}$-UCHL5 rapidly reacted with Ub-VS and Ub$^{I44A,G76V}$-UCHL5, shows a slightly reduced level of activation (Fig. 8d). To further support these findings, we used our *UBE2W* over-expression system to assay UCHL1 activity in whole-cell extracts

using Ub-VS. In this context, the overexpression of *UBE2W* results in the generation of a mixture of UCHL1 and Ub-UCHL1 (Supplementary Fig. 6c, +Dox samples, time 0). Consistent with our previous results, UCHL1 rapidly reacted with the Ub-VS as early as 5 min (Supplementary Fig. 6c, −Dox and +Dox lanes). Of note, there was a small but constant accumulation of Ub$_2$-UCHL1 corresponding to an unspecific interaction of Ub-UCHL1 with Ub-VS. Importantly, Ub-UCHL1 at time 0 was unable to react with the suicide probe as seen by the lack of accumulation of Ub$_2$-UCHL1, suggesting that only the unmodified species of UCHL1 is able to bind and react with the Ub-VS, and not the N-terminally ubiquitinated UCHL1 (Supplementary Fig. 6c). Altogether, these data demonstrate that N-terminal ubiquitination can modulate the deubiquitinase activities of both UCHL1 and UCHL5, but in opposite directions.

Previous reports indicate that UCHL1 DUB activity regulates the free monoubiquitin pool in cells[49]. As we found that N-

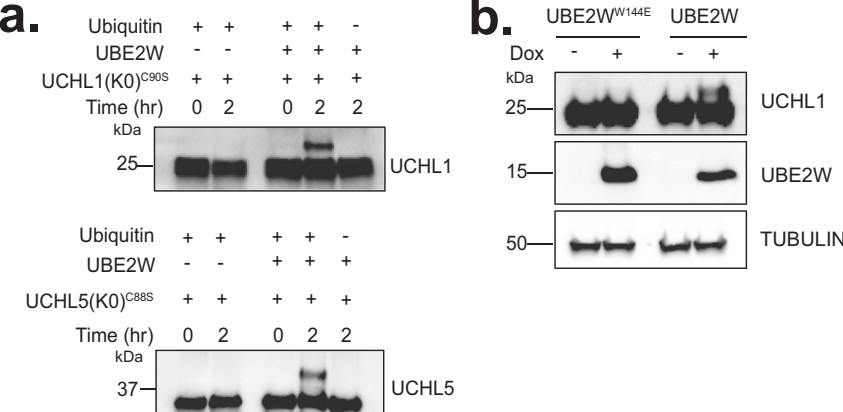

**Fig. 7 UCHL1 and UCHL5 are substrates of UBE2W. a** In vitro ubiquitination assays were performed on lysine-less (K0) catalytically inactive UCHL1$^{C90S}$ (top) and UCHL5$^{C88S}$ (bottom). Two-hour reactions were carried out in the absence of UBE2W (lane 1 and 2) and the presence of UBE2W (lane 3–5). All reactions were incubated with E1, E3 (RNF4), ATP/MgCl$_2$, and with (lane 2) or without (lane 5) ubiquitin. Results are representative of three independent experiments. **b** Western blots of doxycycline-inducible *UBE2W/RNF4* and *UBE2W$^{W144E}$/RNF4* HEK293 cells at 24 h post doxycycline treatment. Endogenous UCHL1 expression was analyzed with an anti-UCHL1 antibody. Results are representative of three independent experiments. Source data are provided as a Source Data file.

terminal ubiquitination negatively regulates UCHL1 activity in vitro, we sought to explore the physiological consequences of this modification in cells. Consistent with previous work in COS-7 cells[44], we found that exogenous expression of *UCHL1$^{WT}$* significantly increased the levels of free monoubiquitin (Fig. 8e, lane 2 compared with lane 1). However, expression of *Ub$^{G76V}$-UCHL1* reduced the accumulation of free monoubiquitin to background levels (Fig. 8e, lane 3). This result supports our in vitro biochemical observations showing that Ub$^{G76V}$-UCHL1 is unable to bind to monoubiquitin (Fig. 8b and Supplementary Fig. 6a). Interestingly, *UCHL1$^{C90S}$* expression triggered the accumulation of free monoubiquitin similarly to *UCHL1$^{WT}$* (Fig. 8e, compare lane 4 with lane 2). However, cells expressing *Ub$^{G76V}$-UCHL1$^{C90S}$* and a non-Ub-binding UCHL1 mutant, *UCHL1$^{D30K}$*, showed basal levels of free monoubiquitin (Fig. 8e, lanes 5 and 6). Collectively, our data indicate that the Ub-binding activity, and not the catalytic activity of UCHL1, regulate the pool of free monoubiquitin in cells. Moreover, these results demonstrate that N-terminal ubiquitination of UCHL1 blocks Ub-binding and inhibits its cellular function.

**N-terminal ubiquitination functions independently of proteasomal degradation.** Although the data demonstrate that UCHL1 is not targeted for proteasomal degradation upon N-terminal ubiquitination, the existing data did not confidently rule out a broader link between N-terminal ubiquitination and proteasomal degradation. To systematically evaluate this connection, we again utilized the *UBE2W* Dox-inducible overexpression model, this time in the presence and absence of proteasome inhibitor Btz. Four individual samples were generated in biological duplicates: the Control (Dox (−)/Btz (−)), Btz proteasome inhibition alone (Dox (−)/Btz (+)), *UBE2W* overexpression alone (Dox (+)/Btz (−)), and Combo (Dox (+)/Btz (+)) (Supplementary Fig. 7). Using the same filtering and cutoffs as for the previous MS experiments, LFQ data for GGX enriched peptides were interrogated to ask whether N-terminal ubiquitination increased upon proteasome inhibition. As in the earlier RNF4 experiments, UBE2W-dependent substrates were identified both in the absence and presence Btz treatment, as represented by the *UBE2W*-Ctrl (Fig. 9a) and combo-Btz (Fig. 9b) contrasts. Interestingly, GGX enriched peptide abundances were systematically unaltered by proteasome inhibition across a wide range of substrates (Fig. 9c

and Supplementary Data 2), confirming our hypothesis that proteasomal degradation is not a major consequence of N-terminal ubiquitination. Consistent with this observation, there is a strong correlation between *UBE2W* overexpressing conditions (i.e., *UBE2W* and Combo), whereas Btz-treated samples closely mirror the controls (Fig. 9d). However, 12 out of the 236 GGX peptides (~5%) did show a coordinate increase in abundance in the combo sample relative to either the Btz treatment alone or *UBE2W* overexpression, as represented by a Log$_2$FC > 2 in both the Combo-*UBE2W* and Combo-Btz contrasts (Supplementary Data 2). Taken together, our label-free proteomic analysis confirms that N-terminal ubiquitination by *UBE2W* is not sufficient to trigger proteasomal degradation on the vast majority of substrates.

**Discussion**

The role of non-canonical ubiquitination, and in particular, N-terminal ubiquitination, remains poorly understood. One challenge has been the identification of enzymes capable of mediating N-terminal ubiquitination. To date, only one Ub-conjugating enzyme, UBE2W[21], attaches Ub to the N-terminus of proteins, and one Ub ligase, the linear Ub chain assembly complex (LUBAC)[22], which attaches Ub to the N-terminus of another Ub molecule, have been described. In addition, it remains unclear which proteins are N-terminally modified and how this Ub impacts protein function and/or stability. Here we have generated a mAb toolkit that is unique relative to existing MS-based methods. We focused on enumerating putative substrates of UBE2W and in so doing, discovered that UBE2W facilitates N-terminal ubiquitination of the UCH DUB family members, UCHL1 and UCHL5. Surprisingly, N-terminal ubiquitination inhibits UCHL1 catalytic activity and increases UCHL5 catalytic activity.

Interestingly, our anti-GGX mAbs identified very few sites of N-terminal ubiquitination that overlapped with sites found by two different strategies that profile global ubiquitination, including N-terminal ubiquitination[19,30]. This result nicely highlights the complementary nature of our mAbs to existing strategies. Cell-specific factors likely have some role in this difference since a majority of sites (60 out of 106) found in a recent study were uniquely observed in the Hep2 cell line, compared with this work which employed HEK293 cells[19]. It remains possible that N-terminal ubiquitination is a stimulus-dependent

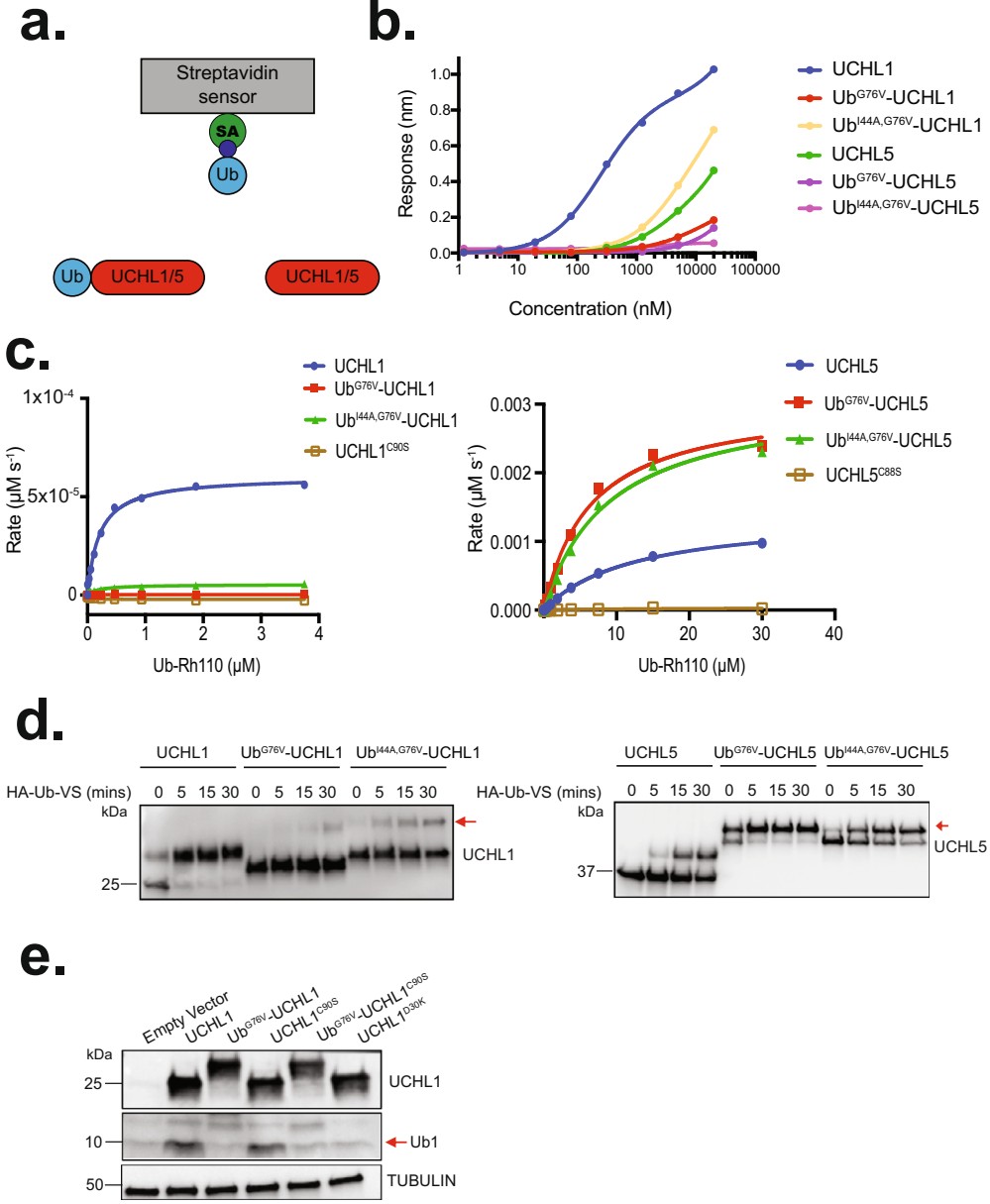

**Fig. 8 N-terminal monoubiquitination of UCHL1 and UCHL5 differentially regulates their DUB activities. a** Graphical representation of the Bio-Layer interferometry (BLI) experiments. UCHL1 and UCHL5 interactions were performed using immobilized biotin-Ub on a streptavidin biosensor, measuring the association of free UCHL1 or UCHL5 and Ub-UCHL1 or UCHL5 with surface-bound Ub. Streptavidin, Ub, and UCHL1/5 are shown in green, blue, and red, respectively. **b** Combined steady-state binding curves for UCHL1, $Ub^{G76V}$-UCHL1, $Ub^{I44A,G76V}$-UCHL1, UCHL5, $Ub^{G76V}$-UCHL5, and $Ub^{I44A,G76V}$-UCHL5 are shown in blue, red, yellow, green, purple, and pink, respectively. Response values in nanometers (nm) are plotted for each concentration of the analyte. Each assay was performed in triplicate. **c** Activity assays were performed with Ub-Rho110 and UCHL1 constructs (left) and UCHL5 constructs (right). For UCHL1 and UCHL5, we included the wild-type protein, catalytically dead mutant ($UCHL1^{C90S}$ or $UCHL5^{C88S}$), or the N-terminally ubiquitinated mimetics ($Ub^{G76V}$ and $Ub^{G76V,I44A}$). UCHL1, $Ub^{G76V}$-UCHL1, $Ub^{I44A,G76V}$-UCHL1, and $UCHL1^{C90S}$ are shown in blue, red, green, and beige, respectively. UCHL5, $Ub^{G76V}$-UCHL5, $Ub^{I44A,G76V}$-UCHL5, and $UCHL5^{C90S}$ are shown in blue, red, green, and beige, respectively. Data are reported as best-fit values with SEs from nonlinear regression fit. Results are representative of two independent experiments. **d** UCHL1 and its N-terminally ubiquitinated mimetic, $Ub^{G76V}$-UCHL1 and $Ub^{I44A,G76V}$-UCHL1 (left) and UCHL5 and its N-terminally ubiquitinated mimetic, $Ub^{G76V}$-UCHL5 and $Ub^{I44A,G76V}$-UCHL5 (right) were allowed to react with the suicide probe ubiquitin-vinyl sulfone (Ub-VS) for the indicated time points. The red arrow indicates the band associated with the N-terminally ubiquitinated mimetic reacting with HA-Ub-VS. Results are representative of three independent experiments. **e** Western blots of COS-7 cells transfected with wild-type UCHL1, $Ub^{G76V}$-UCHL1, $UCHL1^{C90S}$ (catalytically inactive mutant), $Ub^{G76V}$-$UCHL1^{C90S}$, and $UCHL1^{D30K}$ (non-Ub-binding mutant) at 24 h post-Dox treatment. MonoUb is indicated by the arrow. Results are representative of two independent experiments. Source data are provided as a Source Data file.

phenomenon. In similar experiments performed with *Parkin* overexpression, very few substrates are identified in the absence of a ligase activating stimulus—in that case, mitochondrial depolarization[51].

A second factor that likely modulates N-terminal ubiquitination is the availability of free N-termini owing to alterations in N-terminal acetylation. The mechanisms governing N-terminal acetylation remain unclear, although previous studies have found

that cellular stress can significantly alter the extent of N-terminal acetylation, which would, in turn, affect the availability of free N-termini[52,53]. A third factor is the inability of our mAbs to recognize GGP peptides in which proline is positioned immediately adjacent to the N-terminal diglycine (Fig. 1d and Supplementary Fig. 2c). Interestingly, we did identify five internal GGP

peptides, however, this could be explained as non-specific binding of peptides from highly abundant proteins. Almost one-third of previously reported sites of N-terminal ubiquitination occurs at Met-Pro or Pro. Since Met-Pro and Pro cannot be acetylated by N-acetyltransferases (NATs), these sites represent a large pool of free N-termini in cells[54,55]. Proteomics studies also indicate that up to ~60–70% of initiator Met can be removed, suggesting expanding our antibody toolset to other GGX peptides may identify additional N-terminal ubiquitination sites[56,57]. Future work could apply a similar rabbit immune phage method to generate a GGP-specific mAb.

Here, we focused specifically on identifying UBE2W substrates via overexpression owing to low endogenous expression of UBE2W in HEK293 cells. Characterization of UBE2W activity and substrate sequence preference showed that glycines in positions 2–4 of the disordered region increased the enzyme's ability to attach Ub to the N-terminus. In contrast, proline substitutions disrupted Ub attachment to substrate N-termini, suggesting that the amide groups in positions 2–4 are necessary for its recognition[24]. Intriguingly, this result suggests that enzyme(s) other than UBE2W are responsible for a majority of the N-

**Table 1 UCHL1 and UCHL5 kinetics table for Ub-Rho110-monoubiquitin experiment.**

| Protein | $K_m$ ($\mu$M) | $k_{cat}$ ($s^{-1}$) | $k_{cat}/K_m$ ($s^{-1}\mu M^{-1}$) |
|---|---|---|---|
| UCHL1 | 0.2009 | 0.093 | 0.4 |
| $Ub^{G76V}$-UCHL1 | 0.13 | 0.000267 | 0.002 |
| $Ub^{I44A,G76V}$-UCHL1 | 0.198 | 0.005356 | 0.027 |
| $UCHL1^{C90S}$ | ND | ND | ND |
| UCHL5 | 12.38 | 2.792 | 0.22 |
| $Ub^{G76V}$-UCHL5 | 6.395 | 6.112 | 0.95 |
| $Ub^{I44A,G76V}$-UCHL5 | 8.875 | 6.268 | 0.70 |
| $UCHL5^{C88S}$ | ND | ND | ND |

Source data are provided as a Source Data file.

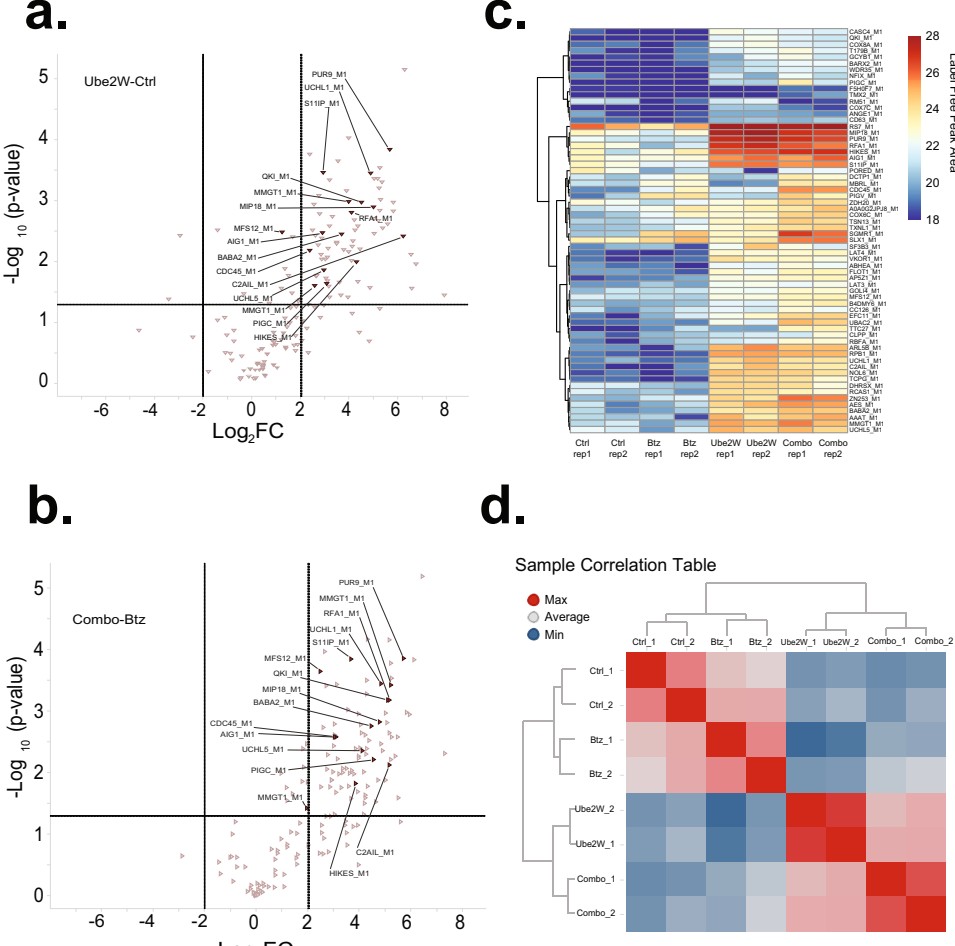

**Fig. 9 N-terminal ubiquitination is unaltered by proteasome inhibition. a, b** Volcano plot showing differential N-terminal protein ubiquitination data for UBE2Woe versus Control (**a**) and Combo versus bortezomib (Btz)-treated (**b**) conditions in label-free GGX-MS experiment. Each data point represents one protein with a representative set of human UniProt proteins shown. **c** Sample correlation table showing Ctrl, Btz-treated, *UBE2W* overexpressing (+Dox) and Combo (*UBE2W* overexpressing + Btz-treated) samples with high correlations shown in red and poorly correlated samples shown in blue. **d** Clustered heatmap of N-terminally modified GGX peptides from selected proteins. Log$_2$-transformed label-free peak areas (AUC) are shown on a color scale ranging from blue to red. Values below 18 were deemed to be indistinguishable from the noise and censored.

terminal ubiquitination sites previously reported, which are enriched for GGP peptides.

UBE2W strictly monoubiquitinates protein substrates at their N-termini[21,23,24], and these modifications can be further modified by the combinate function of E2/E3 complexes into N-terminally linked polyubiquitin chains[23]. Our in vitro ubiquitination assays confirmed that UBE2W is only able to monoubiquitinate substrates (Fig. 7a). However, overexpression studies in cells demonstrated that UBE2W's substrates can be both mono- and polyubiquitinated in cells, confirming previous literature that monoubiquitin on the N-terminus can be further modified to generate polyubiquitinated species (Fig. 5e). Interestingly, expression of UBE2W[W144E], a mutant that lacks the ability to bind to substrates[24], resulted in the absence of the monoubiquitinated forms, but largely maintained the polyubiquitinated species (Fig. 5e). We hypothesize that some substrates may be N-terminally ubiquitinated by UBE2W and elaborated into polyubiquitin chains by other E2 and ubiquitin ligase pairs. Future cellular analysis utilizing UBE2W and catalytically dead mutants of UBE2W and other E2s may provide a better mechanistic view into N-terminally linked polyubiquitin chains and the identification of ubiquitination machinery that cooperates with UBE2W.

UCHL1 is an oncogene, and a recent study showed that high expression of UCHL1 is correlated with worse overall survival in melanoma patients[58]. UCHL1 was previously reported to be N-terminally ubiquitinated[44], and the enzyme responsible for this modification was unknown. Here, we provide biochemical and cellular evidence that UBE2W specifically monoubiquitinates UCHL1 at the N-terminus, resolving a long-standing question regarding the regulation of this DUB[44]. UCHL1 has enzymatic and non-enzymatic roles in regulating the free Ub pool, in part owing to its high monovalent affinity for Ub[48,49]. We confirm that this high monoubiquitin affinity confers a non-enzymatic function to UCHL1 in sequestering monoubiquitin and preventing its incorporation into polyubiquitin chains. We provide further evidence that N-terminal ubiquitination of UCHL1 inhibits its ability to bind Ub and thus inhibits the ability of UCHL1 to regulate the free monoubiquitin pool (Fig. 8b, c, and Supplementary Fig. 6a). In addition, this modification does not promote degradation of UCHL1, nor the vast majority (~95%) of other N-terminally ubiquitinated substrates identified in this study (Fig. 9d and Supplementary Data 2).

UCHL5 is a component of the proteasome 19S regulatory particle and the metazoan INO80 chromatin remodeling complexes[59]. Interestingly, UCHL5 is activated by the proteasome subunit RPN13 and inhibited by the INO80G subunit[59,60]. Strikingly, we observed that UCHL5 is N-terminally ubiquitinated by UBE2W, and this modification significantly enhanced its catalytic activity in vitro (Figs. 7a, 8c, d). The molecular basis of inhibition and activation of UCHL1 and UCHL5, respectively, by N-terminal ubiquitination remains unknown. However, based on the ubiquitin-bound structure of UCHL5[59,61], we hypothesize that since the N-termini of the enzyme is on the opposite side of the crossover loop from the ubiquitin-binding site, the long and flexible crossover loop along with the C-terminal extension in UCHL5, could allow for the N-terminal ubiquitin to occupy the RPN13 site, thus activating the enzyme. In contrast, we observed that the inhibitory role of N-terminal ubiquitination on UCHL1 is in part due to the ability of UCHL1 to bind in cis to its N-terminal Ub modification making the enzyme unable to bind other substrates (Fig. 8b and Supplementary Fig. 6a). Although the binding of Ub[I44A,G76V]-UCHL1 to monoubiquitin was partially rescued, the activity of this mutant remained low (Fig. 8c, d and Table 1). In this context, Ub[I44A,G76V] on the N-terminus of UCHL1 might act as an allosteric inhibitor preventing the re-alignment of the UCHL1 active site residues for enzyme activation. Future

structural analysis may provide a better mechanistic view of activation and inhibition of UCHL5 and UCHL1 by N-terminal ubiquitination, respectively.

In summary, this work establishes an antibody toolkit for global profiling N-terminally ubiquitinated substrates and identifies a role for this non-canonical form of ubiquitination. Here, we have characterized a key enzyme responsible for synthesizing this post-translational modification and provided insight into how substrates are modified at their N-termini. For example, the limited overlap between our UBE2W data set and previously reported N-terminal ubiquitination sites suggest additional enzymes can mediate this post-translational modification, or that biological stimulus might activate or predispose substrates to N-terminal ubiquitination in a conditional manner. Future studies could explore strategies to generate larger potential pools of substrates via modulation of N-terminal acetylation, induction of stress responses, or stimulation of intracellular proteolytic signaling events, such as caspase activation. Furthermore, future studies could further elucidate the enzymatic consequences of N-terminal ubiquitination on both UCHL1 and UCHL5, which is beyond the scope of this work. The availability of complementary strategies to globally profile N-terminal ubiquitination will help clarify the roles of N-terminal ubiquitination in normal physiology and disease processes.

## Methods

**Rabbit immunizations.** Eight New Zealand white rabbits were immunized with a GGM peptide conjugated to either Keyhole limpet hemocyanin (KLH) or ovalbumin (OVA) carrier proteins in order to elicit an immune response in the animal. Rabbits were primed with 500 μg of KLH-linked peptide mixed with Complete freund's adjuvant and subsequently injected intradermally. Four biweekly boosts were done with 250 μg of the peptide antigen in IFA adjuvant. To ensure that we directed the B cell response towards the peptide and not the carrier protein, we alternated the carrier with each boost. After the last boost, 5–10 mL of blood was drawn from each rabbit and protein A-purified pAb sera were generated to monitor the immune response by ELISA. The four rabbits with the best titers were euthanized and the spleen and gut-associated lymphoid tissue (GALT) were harvested. All animals used in this study were housed and maintained at Lampire Biological Laboratories (Pipersville, PA) in accordance with American Association of Laboratory Animal Care guidelines. All experimental studies were conducted under protocols approved by the Institutional Animal Care and Use Committee of Lampire and Genentech Lab Animal Research in an Association for Assessment and Accreditation of Laboratory Animal Care International-accredited facility in accordance with the Guide for the Care and Use of Laboratory Animals and applicable laws and regulations.

**Phage library generation and selections.** Total RNA extracted from the spleen and GALT from four rabbits (rabbits 2, 5, 6, and 7) was used to amplify the variable heavy (VH) and variable light (VL) repertoires separately (Supplementary Table 3). In order to reduce the number of libraries used for selections, we made only four libraries from the total RNA extracted from tissue: (1) rabbits 2 and 6 spleen, (2) rabbits 5 and 7 spleen, (3) rabbits 2 and 6 GALT, and (4) rabbits 5 and 7 GALT. The library diversity for all four libraries was ~1 × 10^8. Using standard Gibson cloning methods, the VH and VL repertoires were assembled into a scFv format and cloned into a phage display vector. A scFv format was chosen because of the improved display compared to a Fab format. The peptide antigens used for selections were either a BSA conjugated or C-terminally biotinylated GGM peptide and a biotinylated K-ε-GG peptide (AAA{K-ε-GG}AAA) for counterselection. Early rounds of selections were carried out using the four individual libraries, but once diversity was reduced, two pooled libraries, the first consisting of the repertoire from rabbit 2 and 6, and the second pooled library containing the repertoire from rabbit 5 and 7, we used for subsequent selections to enable greater sampling of different selections conditions. Three rounds of plate-based selections were done in which bound phage was eluted with 100 mM HCl, neutralized, and amplified in Escherichia coli XL1-blue (Stratagene) with the addition of M13-KO7 helper phage (New England Biolabs). After selections, individual phage clones were picked and grown in 96-well deep-well blocks with 2xYT growth media in the presence of carbenicillin and M13-KO7. After pelleting, the culture supernatants were used in phage ELISAs to screen for specificity.

**pAb ELISAs.** Biotinylated GGM or K-ε-GG peptides were coated at 10 μg/mL in PBS on neutravidin ELISA plates (Thermo Scientific) in triplicate overnight at 4 °C. Plates were washed with PBS with Tween-20 (PBST) solution prior to use. Serial dilutions of protein A-purified pAb starting at 100 μg/mL were incubated for 1–2 h at 25 °C. Plates were washed with PBST. After washing, an anti-rabbit Fc-specific

HRP 2° antibody (Jackson ImmunoResearch, 1:10,000 dilution) was added for 1 h at 25 °C. Plates were washed, developed with TMB substrate for 5 min, and detected at 650 nm.

**mAb ELISAs**. Biotinylated peptides (GGM and K-ε-GG) were coated at 1 μg/mL in PBS on neutravidin ELISA plates (Thermo Scientific) in triplicate overnight at 4 °C. Plates were washed with PBST prior to use. Serial dilutions of anti-GGX mAbs or anti-K-ε-GG mAb (Cell Signaling Technology) starting at 1 μg/mL were added for 1–2 h at 25 °C. Plates were washed and further developed as described above. Biotinylated GGX peptides were synthesized and coated at 1 μg/mL in PBS on neutravidin ELISA plates (Thermo Scientific) in triplicate overnight at 4 °C. Plates were washed and serial dilutions of anti-GGX mAbs starting at 1 μg/mL were added for 1–2 h at 25 °C. ELISA plates were washed with PBST and developed as described above.

**Fab and IgG production**. Constructs for bacterial expression of Fabs were generated by gene synthesis. Fabs were subsequently expressed and purified. In brief, *E. coli* cell paste containing the expressed Fab was harvested from fermentation and dissolved into PBS buffer containing 25 mM ethylenediaminetetraacetic acid (EDTA) and 1 mM PMSF. The mixture was homogenized and then passed twice through a microfluidizer. The suspension was then centrifuged at $21,500 \times g$ for 60 min. The supernatant was then loaded onto a Protein G column (GE Healthcare, Piscataway, NJ) equilibrated with PBS at 5 mL/min. The column was washed with PBS buffer and proteins were then eluted with 0.6% acetic acid. Fractions containing Fabs were pooled and then loaded onto a 50-mL SP Sepharose column (GE Healthcare, Piscataway, NJ) equilibrated in 20 mM MES pH 5.5. The column was washed with 20 mM MES buffer pH 5.5 for two column volumes and then eluted with a linear gradient to 0.5 M NaCl in 20 mM MES buffer pH 5.5. For final purification, Fab-containing fractions from the ion exchange chromatography were concentrated and run on an S75 size exclusion column (GE Healthcare, Piscataway, NJ) in PBS buffer.

Constructs for mammalian expression of rabbit IgGs were generated by gene synthesis. Cells were seeded at $1.0 \times 10^6$ cells/mL for transfection and incubated at 37 °C, 5% CO₂ for 2 h prior to transfection. Plasmids encoding for the LC and HC were diluted in a Dulbecco's modified Eagle's medium (DMEM)-based medium to a final volume of 3 mL. Then 60 μL of 7.5 mM 25 kDa linear PEI (Sigma) was added to the DNA solution, mixed, and incubated at room temperature before being added to the cells. Seven days after transfection, the supernatant was harvested and purified using HiTrap column (GE healthcare) with MabSelect Sure resin (GE Healthcare). PBS was used as the loading buffer. Antibodies were eluted with 0.1 M citrate, pH 3.0, and neutralized with 3 M Tris, pH 8.0, to a final pH of 7.0. Each antibody was further polished via size exclusion chromatography to remove any aggregates and increase the homogeneity of monomeric antibodies to at least 95%. Samples were run on a Superdex S200 10/300 GL size exclusion column (GE Healthcare) using PBS load buffer at a flow rate of 1 mL/min (30 cm/h). Pooled fractions were filtered using a 0.2 μm filter, and antibody monomer content of the final antibody preparation was assessed by analytical SEC carried out with a TSK-GEL, Super SW3000, 4.6 mm × 30 cm, 4 μm (Tosoh Bioscience) column using a Dionex Ultimate 3000 system (Thermo Fisher Scientific) at a flow rate of 0.3 mL/min.

**Crystallization conditions and structure determination**. The Fab-GGM complex was screened for crystallization using the hanging drop method with 1:1 ratio of protein:well-solution. Crystals were observed in multiple conditions, with the best condition being 2 M ammonium sulfate and 0.1 M TRIS pH 7.5. Upon optimization, single crystals grew to ~200 mm in 2 M ammonium sulfate and 0.1 M MES pH 6.5. The crystals matured over 2 weeks and were flash-frozen with 20% (v/v) ethylene glycol in 2 M ammonium sulfate and 0.1 M MES pH 6.5. The diffraction data were collected at advanced light source (ALS) beamline 5.0.2 at a temperature of 100 K. The data were processed to 2.85 Å resolution with HKL2000[62] and the phases were obtained with PHENIX by molecular replacement with a model rabbit Fab (PDB: 4ZTP)[63]. The structure was built using COOT[64] and refined using PHENIX[65]. The final model was generated after the addition of GGM peptides, water molecules, and buffer molecules (Supplementary Table 1).

**Pilot MS experiments demonstrating reagent selectivity**. In all, 40 mg of protein lysate was prepared from confluent HEK293T cells and digested with Trypsin (Promega). Tryptic peptides in PTMScan® IAP buffer (Cell Signaling Technologies) were incubated with 80 μg of anti-GGX or anti-K-ε-GG mAb (Cell Signaling Technology) for 30 min at 4°C. Subsequently, 80 μL of protein G agarose slurry was added to the antibody-peptide mixture for an additional 30 min at 4°C. For the MS experiment in which the four anti-GGX mAbs were pooled for use in peptide immunoaffinity enrichment, 50 μg of each mAb was mixed together before being put into contact with trypsin digested and desalted peptides.

**MS Analysis of Pilot GGX experiments**. Resuspended samples were analyzed by LC-MS/MS in a 120 min total run-time method. Peptides were separated on a nanoAcquity UPLC (Waters) and introduced to either an Orbitrap Elite or Q Exactive HF mass spectrometer (ThermoFisher) by electrospray ionization. Thirty to forty percent of each sample was loaded onto a 100 μm × 100 mm Waters 1.7-μm

BEH-130 C18 column and separated by low pH reversed-phase chromatography (solvent A: 0.1% FA/98% water/2% acetonitrile (ACN), solvent B: 0.1% FA/98% ACN/2% water) at a flow rate of 1 μl/min using a two-stage linear gradient applied over 90 min. In the first stage, solvent B increased from 2% to 25% over a span of 85 min, followed by the second stage with solvent B increasing from 25% to 40% over a span of 5 min. Both the Orbitrap Elite and Q Exactive HF MSs were operated in data-dependent mode with the top 15 and top 10 most abundant ions selected for MS2 fragmentation, respectively. MS-specific settings used for analysis, optimized for each instrument, were as follows.

Orbitrap Elite FTMS1 scans were collected at 60,000 resolution, an AGC target of $1 \times 10^6$, and a maximum injection time of 200 ms. ITMS2 was performed using CID set at 35% normalized collision energy, an AGC target of $1 \times 10^3$, and a maximum injection time of 100 ms.

Q Exactive HF FTMS1 scans were collected at 60,000 resolution, an AGC target of $3 \times 10^6$, and a maximum injection time of 60 ms. FTMS2 was performed using HCD set at 30% normalized collision energy, collected at 15,000 resolution, an AGC target of $1 \times 10^5$, and a maximum injection time of 75 ms.

**Data analysis**. Data files were searched using Mascot (Matrix Science) against a target-decoy database containing Uniprot human (downloaded August, 2017) and common contaminant sequences. A precursor ion mass tolerance of 25 ppm, fragment ion tolerance of 0.8 Da (ITMS2) or 0.02 Da (FTMS2), and semi-tryptic enzyme specificity were employed. Carbamidomethylated cysteine (+57.0215 Da) was set as a fixed modification and methionine oxidation (+15.9949 Da), K-ε-GG (+114.0429), and N-terminal GG (+114.0429) were considered as variable modifications. PSMs were filtered at the peptide level using linear discriminate analysis to a false discovery rate of 5% with subsequent filtering based on sequence features relevant to the biology being interrogated. Peptide tandem mass spectra were plotted and visualized using the web-based spectrum annotator, Interactive Peptide Spectral Annotator[66]. We noted during data analysis that a subset of matching PSMs came from protein fragments present in the database as unreviewed TrEMBL entries with a variable 'X' as the first amino acid. These PSMs were eliminated from consideration as possible N-terminal ubiquitinated species.

**Immunoaffinity enrichment of GGX and K-ε-GG peptides from *UBE2W* expressing cells for LFQ analysis by MS**. In all, $7 \times 15$ cm plates of HEK293 cells inducibly expressing *UBE2W* and matched controls were lysed under fully denaturing conditions (9 M urea, 20 mM HEPES pH 8.0, 1 mM sodium orthovanadate, 2.5 mM sodium pyrophosphate, 1 mM β-glycerophosphate). Lysates were microtip sonicated on ice ($2 \times 30$ sec), and cleared by high-speed ultracentrifugation ($18,000 \times g$, 15 min). In all, 40 mg of each lysate was taken forward for reduction (4.1 mM dithiothreitol, 60 min at 37°C), alkylation (9.1 mM iodoacetamide, 15 min at room temperature), fourfold dilution, and then subjected to overnight digestion with a combination of lysyl-endopeptidase (Wako) and sequencing grade trypsin (Promega) both at an enzyme to protein ratio of 1:100, the latter which was added 4 hr following incubation with the former. Digested peptides were acidified with TFA to a final concentration of 1%, cleared by centrifugation ($18,000 \times g$, 15 min), desalted by Sep-Pak C18 gravity flow solid-phase extraction (Waters), and lyophilized for 48 h. Dry peptides were reconstituted in 1 mL 1× IAP buffer (Cell Signaling Technology) and clarified via high-speed centrifugation ($18,000 \times g$, 10 min) for subsequent immunoaffinity enrichment.

Peptides were subjected to two serial rounds of immunoaffinity enrichment, both performed at 4 °C on a MEA2 automated purification system (Phynexus) using 1 mL Phytips (Phenexus) packed with 20 μL ProPlus resin coupled to either 200 μg of anti-GGX antibody cocktail or 200 μg of anti-K-ε-GG (Cell Signaling Technology) antibody. Enrichment was performed in the following order: anti-GGX IP for peptides containing a diglycine-modified N-terminus (GGX), and anti-K-ε-GG IP for peptides containing diglycine-modified lysine residues (K-ε-GG). For the experiment involving *UBE2W* overexpression and Btz treatment, K-ε-GG was not performed following GGX peptide enrichment.

Immunoaffinity enrichment on the MEA2 was performed[67]. In brief, Phytip columns were equilibrated for two cycles (1 cycle = aspiration and dispensing, 0.9 mL, 0.5 mL/min) with 1 mL 1× IAP buffer prior to contact with peptides, incubated with peptides for 16 cycles of capture, and washed for six cycles (2× with 1 mL 1× IAP buffer followed by 4× with 1 mL water). Captured peptides were eluted with 60 μL 0.15% TFA in eight cycles where the volume aspirated/dispensed was adjusted to 0.06 mL. Eluted peptides were subsequently desalted using C18 stage-tips[68] and SpeedVac (ThermoFisher) dried to completion.

Enriched GGX peptides were reconstituted in 2% acetonitrile/0.1% formic acid (FA) and analyzed in duplicate (40% each injection) by LC-MS/MS on an Orbitrap Lumos mass spectrometer (ThermoFisher) coupled to a Dionex Ultimate 3000 RSLC (ThermoFisher) employing a 100 μm × 250 mm PicoFrit (New Objective) column packed with 1.7-μm BEH-130 C18 resin (Waters). Low pH reversed-phase separation (solvent A: 0.1% FA/98% water/2% ACN, solvent B: 0.1% FA/98% ACN/2% water) was performed at 450 nL/min on a 96 min two-step linear gradient where solvent B increased from 2% to 35% over 102 min and then from 35% to 50% over 2 min with a total run time of 120 min. The Orbitrap Lumos was operated in a data-dependent mode whereby FTMS1 scans were collected at 240,000 resolution with an AGC target of $1 \times 10^6$ and a maximum injection time of 50 ms. MS2 scans on the top 15 most intense precursors with charge states of 2–4

were collected in the ion trap with HCD fragmentation at a normalized collision energy of 30%, an AGC target of $2.0 \times 10^4$, and a max injection time of 11 ms.

For the duplicate injections, MS2 spectra were analyzed in the Orbitrap rather than the ion trap. OTMS2 AGC target was set to $2.0 \times 10^5$ with a max injection time of 54 ms.

**Sample correlation**. The Sample Correlation heatmap was prepared in R using the *stats* package. Pearson correlation coefficients were calculated pairwise between all samples using the "cor" function. Correlation scores between biological replicates were computed using each protein's model abundance in the respective replicates.

**Heatmap**. The Label-Free Peak Area heatmap was prepared using the R package *pheatmap*. Proteins were clustered row-wise into six clusters using protein model abundance values. Clusters 2, 3, and 5, identified as those showing patterns consistent with the proteins being substrates of UBE2W, were selected and plotted as a heatmap in a long format using protein model abundance values.

**Immunoaffinity enrichment of GGX and K-ε-GG peptides from *UBE2W* and/or *RNF4* expressing cells for LC-MS analysis**. Immunoaffinity enrichment of GGX and K-ε-GG peptides from 40 mg each of HEK293 cells either noninduced ($N = 3$) or inducibly expressing *RNF4* ($N = 2$), *UBE2W* ($N = 3$), or the combination ($N = 3$) was performed as detailed above with the following modifications.

Peptides were subjected to three serial rounds of immunoaffinity enrichment, all performed as described above on a MEA2-automated purification system (Phynexus) with either 200 μg of anti-GGX antibody cocktail or 200 μg of anti-K-ε-GG (Cell Signaling Technology) antibody. Enrichment was performed in the following order: anti-GGX IP for peptides containing a GGX, anti-K-ε-GG IP, followed by anti-GGX IP for GGX once more.

The enriched peptides from first (GGX) and second (K-ε-GG) round immunoprecipitations were subsequently prepared for TMT-11 multiplexed quantitative analysis[42,67] while the enriched GGX peptides from the third round were prepared for LFQ MS analysis.

**TMT-11 multiplexed sample prep**. Eluates containing enriched GGX or K-ε-GG peptides were desalted using C18 stage-tips, SpeedVac dried to completion, and reconstituted in 25 μL 200 mM HEPES pH 8.0 for subsequent isobaric labeling with 11-plex TMT reagents (ThermoFisher). Each vial of TMT reagent was allowed to thaw for 5 min at room temperature, spun down using a benchtop centrifuge, and resuspended in 41 μL of anhydrous acetonitrile (ACN). To each eluate, 8 μL of TMT reagent was added along with 2 μL of ACN to reach an optimal labeling reaction final ACN concentration of 29%. After 1 hr incubation at room temperature, the reaction was quenched by the addition of 4 μL of 5% hydroxylamine for 15 min. Labeled peptides were combined and dried by vacuum centrifugation.

The TMT labeled GGX peptides were resuspended in solvent A (2% ACN/0.1% formic acid (FA)) and split into two portions, 40% and 60%, the former slated for LC-MS/MS analysis without further manipulation and the latter subjected to additional offline high pH reversed-phase fractionation using an RPS cartridge on the AssayMap (Agilent) employing a 0.1% triethylamine/acetonitrile based elution buffer. Six fractions were collected (F1: 12% ACN, F2: 17% ACN, F3: 22% ACN, F4: 27% ACN, F5: 32% ACN, F6: 80% ACN). Fractionated GGX peptides were subsequently lyophilized and resuspended in solvent A for LC-MS/MS analysis.

For TMT labeled K-ε-GG peptides, high pH reversed-phase fractionation was performed using a commercially available kit (ThermoFisher). Upon resolubilization in 0.15% TFA, fractionation was performed according to the manufacturer's protocol with a modified elution scheme where 11 fractions were collected (F1: 13.5% ACN, F2: 15% ACN, F3: 16.25% ACN, F4: 17.5 ACN, F5: 20% ACN, F6: 21.5% ACN, F7: 22.5% ACN, F8: 23.75% ACN, F9: 25% ACN, F10: 27.5% ACN and F11: 30% ACN) and then combined into six fractions (F1 + F6, F2 + F7, F8, F3 + F9, F4 + F10, F5 + F11). Peptides were lyophilized and resuspended in 10 μL solvent A for LC-MS/MS analysis.

LC-MS/MS analysis on the unfractionated GGX sample was performed on an Orbitrap Lumos mass spectrometer (ThermoFisher) coupled to a NanoAcquity UPLC (Waters) system equipped with a 100 μm × 250 mm PicoFrit column (New Objective) packed with 1.7 μm BEH-130 C18 (Waters). Low pH reversed-phase separation (solvent A: 0.1% FA/98% water/2% ACN, solvent B: 0.1% FA/98% ACN/2% water) was performed at 500 nL/min on a 163 min two-step linear gradient where solvent B was ramped from 2% to 30% over 158 min and then from 30% to 75% over 5 min with a total run time of 180 min. The Orbitrap Lumos collected FTMS1 scans at 120,000 resolution with an AGC target of $1 \times 10^6$ and a maximum injection time of 50 ms. FTMS2 scans on precursors with charge states of 2–6 were collected at 15,000 resolution with CID fragmentation at a normalized collision energy of 35%, an AGC target of $5.0 \times 10^4$, and a max injection time of 200 ms. Synchronous-precursor-selection MS3 scans were analyzed in the Orbitrap at 50,000 resolution with the top eight most intense ions in the MS2 spectrum subjected to HCD fragmentation at a normalized collision energy of 55%, an AGC target of $1.5 \times 10^5$, and a max injection time of 400 ms.

LC-MS/MS analysis on the fractionated GGX peptides was performed as described above with the following exceptions. Liquid chromatography was performed using a Dionex Ultimate 3000 RSLC (ThermoFisher) on an Aurora

Series 25 cm × 75 μm I.D. column (IonOpticks) running at a reduced flow rate of 300 nL/min and a modified gradient whereby solvent B ramped from 2% to 30% over 135 min and 30% to 50% over 15 min.

LC-MS/MS analysis on the fractionated K-ε-GG peptides was performed exactly as described for the unfractionated GGX sample with a modification to the MS method restricting precursor ions selected for fragmentation to those with charge states 3–6.

In the TMT analysis, a series of large and unanticipated features were observed in the MS1 data that appeared to affect the performance of the data-dependent method. These features included a series of intense peaks eluting across the chromatogram, which were identified as the compounded signal of abundant, internal GGX peptides present in each of the 11 samples. We hypothesized that this signal overshadowed lower intensity signals from N-terminal ubiquitinated GGX peptides of interest that are expected to be present in only a subset (*UBE2W* only, Combo) of the 11 samples. As with many LC-MS experiments, we acquired data using an approach referred to as data-dependent sampling where the instrument is directed to isolate and fragment the most abundant peptide ions entering the instrument at any moment in time. In this instance, the UCHL1 GGX peptides did not quite make their way onto the list of features analyzed by MS2, and thus there is an absence of evidence for these features. This absence of evidence should not be construed as evidence of absence given the nature of data-dependent shotgun sequencing. In an effort to minimize competition of high abundance internal GGX peptides for signal and recover additional identifications that may have been obscured, immunoaffinity enrichment using the flowthrough peptides from the TMT labeling experiment and subjected those enriched peptides to LC-MS for LFQ analysis, as above. For TMT multiplexing data, raw MS data were searched using Mascot against a Uniprot human target-decoy database (downloaded August, 2017) containing common contaminant sequences with a ppm precursor ion mass tolerance of 25 ppm, fragment ion tolerance of 0.02 Da, and semi-tryptic enzyme specificity. Carbamidomethylated cysteine (+57.0215 Da) and TMT labeled N-termini (+229.1629) were set as fixed modifications with methionine oxidation (+15.9949 Da), K-ε-GG (+114.0429), and N-terminal GG (+114.0429) considered as variable modifications. PSMs for each run were filtered using LDA to an FDR of 3% and a ppm mass tolerance between −5 and 4 from the theoretical precursor m/z. TMT-MS3 quantification was performed using Mojave[69]. Quantification and statistical testing of the TMT proteomics data were performed using MSstatsTMT v1.6.3, an open-source R/Bioconductor package[70]. Prior to MSstatsTMT analysis, PSMs were filtered from further analysis if they were (1) from decoy proteins; (2) from peptides with length <7; (3) with isolation specificity <50%; (4) with reporter ion intensity less than 256; or (5) with summed reporter ion intensity (across all eleven channels) lower than 30,000. Redundant PSMs (i.e., multiple PSMs in one MS run that map to the same peptide) were summarized by first taking the maximum reporter ion intensities per peptide and channel and then selecting the fraction with the maximum reporter ion intensity for each PSM. Next, MSstatsTMT summarized the peptides to the protein modification site level using Tukey median polish summarization (TMP). The differential abundance analysis between conditions was calculated by MSstatsTMT based on a linear mixed-effects model per protein. The inference procedure was adjusted by applying an empirical Bayes shrinkage, and resulting $p$ values adjusted for multiple hypothesis testing by the Benjamini–Hochberg procedure.

In the pooled TMT samples, we observed that internal GGX peptides displayed disproportionately high signals relative to the N-terminally ubiquitinated GGX remnants. In an effort to overcome this effect and capture additional UBE2W substrates, additional immunoaffinity enrichment experiments and label-free MS analyses were performed on the Control (no dox), *UBE2W* only, *RNF4* only, and *RNF4/UBE2W* (Combo) samples.

LC-MS/MS was performed similarly to the two conditions LFQ experiment with the following minor modifications to liquid chromatography and data acquisition. Low pH reversed-phase separation was performed at a flow rate of 450 nL/min on an Aurora Series 25 cm × um I.D. column (IonOpticks). The dual-stage gradient was modified to ramp from 2 to 35% solvent B over 91 min and 35 to 75% over 5 min. For all injections, MS2 spectra were analyzed in the ion trap.

These MS data were searched using Mascot (Matrix Science) against a target-decoy database (downloaded August, 2017) containing Uniprot human and common contaminant sequences using a ppm precursor ion mass tolerance of 25 ppm, fragment ion tolerance of 0.8 Da, and semi-tryptic enzyme specificity. Carbamidomethylated cysteine (+57.0215 Da) was set as a fixed modification and methionine oxidation (+15.9949 Da), K-ε-GG (+114.0429), and N-terminal GG (+114.0429) considered as variable modifications. PSMs were filtered at the peptide level using linear discriminate analysis at a false discovery rate of three percent. LFQ of the N-terminal GG peptides across all datafiles was performed using XQuant, an algorithm guided by direct PSMs that utilizes accurate precursor ion masses and retention times to quantify peptides across runs[71]. Quantification and statistical testing of the label-free proteomics data were performed using MSstats v3.20.0, an open-source R/Bioconductor package[72]. Prior to MSstats analysis, PSMs were removed from further analysis if they were (1) from decoy proteins; (2) from peptides with length <7; (3) possessed VistaQuant confidence scores <71; or (4) with peak area <256. Redundant PSMs (i.e., multiple PSMs in one MS run that map to the same peptide) were summarized by taking the maximum intensities per run. Next, MSstats summarized the peptides to the protein modification site level using TMP. The differential abundance analysis

between conditions was calculated by MSstats based on a linear mixed-effects model per protein. P values from the linear mixed-effects model were adjusted for multiple hypothesis testing by using the Benjamini–Hochberg procedure. PSMs corresponding to TrEMBL entries reporting 'X' as the first amino acid was removed from consideration as N-terminally ubiquitinated species following quantitative analysis.

**Cell culture.** HEK293 and COS-7 cell lines were obtained from Genentech's cell line core facility gCell. Cells were maintained in DMEM supplemented with 10% FBS, 2 mM L-glutamine, and 50 U/ml penicillin–streptomycin. All cell lines were cultured in a humidified incubator at 37 °C/5% $CO_2$ and media were changed every other day. If applicable, cells were treated with vehicle dimethyl sulfoxide (Cat# D2650, Sigma-Aldrich), 1 µg/ml Dox (Cat# D9891, Sigma-Aldrich), 1 µM Btz (Cat# 2204, CST), and 10 µg/ml cycloheximide (Cat# 2112, CST) for the indicated times.

**DNA constructs, transfection, and western blotting.** All DNA constructs were obtained by custom gene synthesis (GeneScript) and subcloned into the Dox-inducible piggyBac transposon plasmids (BH1.2, Genentech) using the NcoI and XhoI sites. For transient expression, HEK293 and COS-7 cells were seeded in six-well plates and grown to ~50% confluence in DMEM media. The cells were then transfected with 1 µg of piggyBac transposon plasmid by using 10 µl Fugene (Promega) according to the manufacturer's instructions. For stable cell line generation, cells were cotransfected with 250 ng of piggyBac transposase plasmid (Transposagen) and 750 ng of the piggyBac transposon plasmid, using 10 µl of Fugene (Promega). Three days after transfection, cells were split into selection media containing 1 µg/mL puromycin and selected for 10 days.

Stable or transient transfected cells were then assayed for protein expression by western blot analysis. Two days after treatment with 1 µg/mL Dox, cells were lysed with denaturing lysis buffer (9 M urea, 20 mM HEPES pH 8.0), sonicated, and centrifuged at 16,000 × g for 10 min at 4 °C. In all, 15–50 µg of protein was prepared in 1× SDS loading buffer (ThermoFisher) and 1× reducing agent (ThermoFisher), heated to 90 °C for 5 min, and ran in 12% Tris-glycine gels (Bio-Rad). Gels were transferred to a nitrocellulose membrane using a Trans-Blot Turbo System (Bio-Rad) for 7 min at 23 V. Membranes were blocked with 5% nonfat milk diluted in PBST for 30 min and were rinsed briefly three times with PBST and incubated overnight at 4 °C with primary antibodies in PBST with 5% BSA. Blots were washed three times for 5 min in PBST and then incubated for 1 h at room temperature with secondary antibodies in PBST with 5% BSA. Blots were washed and detection was performed with Supersignal Femto (Pierce). The antibodies were 1:5000 rabbit anti-beta Tubulin (Cat# ab6046; Abcam), 1:1000 rabbit anti-UBE2W (Cat# PA5-67547; Thermo Fisher),1:2000 rabbit anti-HA tag (Cat#3724 S, Cell Signaling Technology), 1:1000 mouse anti-RNF4 (Cat#MA527423, Thermo Fisher), 1:1000 rabbit anti-p21 (Cat#ab109199, Abcam), 1:2000 rabbit anti-UCHL1 (Cat# HPA005993; Thermo Fisher), 1:2000 rabbit anti-UCH37 (Cat# ab124931; Abcam), 1:500 mouse anti-ubiquitin (Cat# VU-1; LifeSensors), and 1:10,000 goat anti-mouse and rabbit IgG HRP (Cat# 31460 and 31430, Thermo Fisher, respectively).

**Ubiquitination assays.** For ubiquitination assays, we used a mix of 100 nM E1 (Cat# E-305, Boston Biochem), 4 µM UBE2W (Cat# E2-740, Boston Biochem), 1 µM UCHL1-K0 and UCHL5-K0 (made in house), 1 µM RNF4 (Cat# E3-210, Boston Biochem), and 250 µM Ub (Cat# U-100H, Boston Biochem). All reactions were performed in 40 µl ubiquitination buffer (50 mM tris pH 7.5, 5 mM MgCl2, 50 mM KCL, and 0.2 mM DTT) at 37 °C for 2 h. Reactions were started with 3 mM ATP and stopped by the addition of Laemmli buffer and heated to 90 °C, followed by separation of proteins by SDS-PAGE and visualization by immunoblotting with the appropriate antibodies.

**Ub-Rhodamine 110 enzymatic assays.** Ub-Rhodamine 110 (Cat# SBB-PS0001, South Bay Bio) activity assays were determined using 1 nM of purified enzyme with increasing concentration of substrate (Ub-Rho110) in 10 µl reaction buffer (50 mM HEPES pH 7.5, 50 mM KCl, 5% glycerol, 5 mM MgCl2, 5 mM DTT, 0.1 mg/ml BSA, and 0.005% Tween-20). Experiments were performed at 37 °C in black 384-well non-binding surface low flange plates (Corning) and monitored in an EnVision 2105 Multimode Plate Reader (PerkinElmer) using 350 nm and 450 nm excitation and emission wavelengths, respectively. Measurements were taken every 60 s for 90 min.

**Ubiquitin-vinyl sulfone assays.** Purified proteins (50 nM) were subjected to enzymatic reactions with 1 µM Ub vinyl sulfone HA-tagged probe (Ub-VS-HA) (Cat #U-212, Boston Biochem). All reactions were done in 40 µl of DUB buffer (50 mM HEPES pH 7.5, 50 mM KCl, 5% glycerol, 5 mM MgCl2, 5 mM DTT, 0.1 mg/ml BSA, and 0.005% Tween-20) at 37 °C for 30 min. Modification of enzymes by site-directed HA-Ub-VS probes was detected by immunoblotting with the appropriate antibodies.

For Ub-VS assays with whole-cell extracts, (Dox)-inducible *UBE2W* HEK293 cells were grown in 15 cm plates and incubated with and without Dox for 12 h. Cells were then harvested with a cell scraper and resuspended with 1 ml of lysis buffer (50 mM Tris pH 8, 50 mM NaCl, 1 mM EGTA, 1 mM EDTA, 5 mM MgCl2,

2 mM ATP, 5 mM tris(2-carboxyethyl)phosphine (TCEP), 250 mM sucrose, 5% glycerol). Lysates were then sonicated on ice (2 × 30 sec), and centrifuged at max speed for 10 min. The supernatant was transferred to a new tube and 200 µg of total protein was used in 40 µl reactions with 1 µM Ub-VS. Reactions were performed at 37 °C and 10 µl aliquots were taken at the indicated time points.

**Protein expression and purification.** All full-length WT and mutant proteins, as well as N-terminally fused Ub, were obtained by custom gene synthesis (GeneScript) and subcloned into single protein expression vectors for expression in *E. coli*. All sequences were tagged at their C-termini with 6-His tags. Proteins were expressed in BL21-Gold(DE3) cells for 18 h at 18 °C, then harvested by centrifugation in lysis buffer containing 500 mM NaCl, 50 mM Tris 7.5, 5% glycerol, and 1 mM TCEP. Proteins were purified by affinity chromatography (Ni-NTA Agarose, Thermofisher) followed by size exclusion chromatography (16/600 Superdex200, GE Healthcare). Protein samples were concentrated and frozen down in GF Buffer (150 mM NaCl, 20 mM Tris 7.5, 1 mM TCEP).

**BLI assay.** BLI assays were run on Octet Red384 (Forte Bio) platform using 384 tilted-well plates. All experiments were conducted at 25 °C, 1000 RPM shaking, 60 µL well volume, and used buffer containing: 150 mM NaCl, 20 mM Tris 7.5, 1 mg/mL BSA, 0.01% Tween-20, and 1 mM TCEP. 60 s baseline or wash steps preceded each loading, association, or dissociation step in a blank buffer. Immobilization of biotinylated ubiquitin (Cat# UB-570, Boston Biochem) was optimized on Streptavidin biosensors (Cat# 18-5019, Forte Bio) to 18 nM (0.156 µg/µL) for a 1 nM response over a 300 s loading step. The Association of the wild-type or ubiquitin-fused proteins was carried out a fourfold dilution of protein starting at 5 µM. The association step was measured for 180 s, and the dissociation step was measured for 300 s. Streptavidin tips with no biotin-ubiquitin loaded were used to measure the non-specific binding of each protein dilution with the tip surface, and the data was subtracted from the raw data measurements before curve fitting. Association and dissociation curves were fitted using a 1:1 binding model in the prism, with $K_D$ being determined from kinetic constants as well as steady-state measurements.

**Reporting summary.** Further information on research design is available in the Nature Research Reporting Summary linked to this article.

## Data availability
The existing rabbit Fab structure used as a model for molecular replacement is available via protein databank (PDB) accession code 4ZTP. The new crystal structure has been deposited with the PDB accession code 7MFR. The MS raw files have been uploaded to the UCSD MassIVE repository with accession code MSV000086537. The Uniprot (https://www.uniprot.org) and TrEMBL databases (https://www.uniprot.org/uniprot/?query=reviewed:no) were used for the MS analysis. Source data are provided with this paper.

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

## Acknowledgements

PTMscan® has been performed at Genentech under a limited license from Cell Signaling Technologies. Figures where applicable were created with BioRender.com. We thank members of the antibody production and automation technologies (APAT) group for help in antibody and Fab purification.

## Author contributions

C.W.D., S.E.V., D.S.K., I.E.W., and J.T.K. conceived the project. C.W.D. generated and characterized the rabbit antibodies. J.S. generated crystals and solved the crystal structure. L.P., T.B.H., F.R.S., Y.J.Z., J.R.L., C.M.R., and D.S.K. designed, performed, and analyzed the MS data. S.E.V., S.C.R., A.S.P., A.S.S., and I.E.W. designed, performed, and analyzed the western blot experiments to validate substrates and biochemical experiments to characterize UCHL1 and UCHL5 activities. C.S. prepared Ub-UCHL1 and UCHL5 proteins and provided feedback on experiments. C.W.D., S.E.V., D.S.K., and J.T.K. wrote the paper with input from all of the authors.

## Competing interests

C.W.D., S.E.V., L.P., J.S., T.B.H., S.C.R., J.R.L., C.M.R., A.S.S., and J.T.K. are current employees of Genentech Inc. F.R.S., Y.J.Z., A.S.P., I.E.W., and D.S.K. are former employees of Genentech Inc. I.E.W. is a current employee of Bristol Myers Squibb Inc. D. S.K. is a current employee of Interline Therapeutics. C.S. is a current employee of Boston Biochem, Inc.
