## [Peer Review File · Nature Communications]

REVIEWER COMMENTS

Reviewer #1 (Remarks to the Author):

The authors present novel antibodies for the enrichment of N-terminally ubiquitinated substrates that they developed by combining the use of Gly-Gly-Met (GGM) peptide as antigen for rabbit immunizations with phage display. They made sure that these antibodies have minimal cross reactivity with peptides bearing K-ε-GG motif classically found in Lys-modified ubiquitination substrates and thoroughly validated their affinity to other GGX peptides.

1. The authors generated antibodies against GGM peptide that can also interact with a subset of GGX peptides. As GGM peptide is also a signature peptide for linear ubiquitin linkages, these antibodies could also enrich such Ub chains (either generated by LUBAC or nascent Ub chains upon translation of tandem/polyubiquitin chains) – please provide data that show how these antibodies recognize linear linkages (under conditions favoring generation of linear ubiquitin chains) and nascent polyubiquitin chains.
2. Can authors validate putative UBE2W N-terminal ubiquitin-modified substrates also in UBE2W KO cells? Is any N-terminal ubiquitination observed in that context?
3. Figure 4E. Authors state that identified UBE2W substrates show enrichment of the monoubiquitinated form, with certain proteins being modified by polyubiquitin chains. However, in most of the examples in Figure 4E there is a clear difference between UBE2W-overexpressed cells (WT or W144E) and cells only having low endogenous UBE2W levels. However, there is no striking difference between UBE2W wt and mutant cells, as even mutant cells have often smear above substrate. How can authors explain that? How does C91S mutant of UBE2W behave in such validation?
4. As authors could not observe modification of endogenous UCHL5 in WB, were they able to find the modification of endogenous UCHL5 by MS approaches?
5. Does Ile44 mutant of Ub fused to N-terminus of UCHL1 or UCHL5 affect in the same way as wild-type Ub the catalytic activity and Ub-binding properties of UCHL1 and UCHL5? Is UCHL1 able to interact in cis with its N-terminal Ub modification? In other words, does the ability of UCHL1/UCHL5 to bind Ub directly affect catalytic activity of these enzymes?
6. Is UCHL5 recruitment to 19S proteasome and binding to Rpn13 affected by N-terminal modification?

Minor comments

1. To which Institution do all the authors (except 5: Carsten Schwerdtfeger) belong to? Only department names are provided. I assume Genentech?
2. Word(s) is missing in the sentence: "Subsequent work demonstrated that N-terminally ubiquitinated proteins do not accumulate significantly upon proteasome inhibition, suggesting that N-terminal ubiquitination might have additional roles beyond promoting proteasome-mediated degradation¹⁹, such as serving as (word missing) assisting in the folding of nascent polypeptides²⁰
3. Discussion sentence "To date, only one Ub-conjugating enzyme, UBE2W²¹, and one Ub ligase, the linear Ub chain assembly complex (LUBAC)²², have been described to attach Ub or Ub chains at the N-terminus, respectively. " is a bit misleading, as it suggests that LUBAC can attach Ub chains to a subset of substrates. As far as reviewer knows, LUBAC has only been shown to attach Ub to another Ub molecule, in head-to-tail mode, through peptide bond between Gly76 residue of one Ub and Met1 residue of another Ub. Please rephrase for clarity.
4. Figure 4E. Labeling for HA has been partially cropped in panels 1 and 3.

Reviewer #2 (Remarks to the Author):

J Koerber and coworkers studied N-terminal ubiquitination by UBE2W through using a novel antibody toolkit. They identify 4 antibodies that selectively recognize tryptic peptides Gly-Gly-X (ubiquitination site) but not ubiquitin modifications on lysine and solve the structure of one of these antibodies bound to a Gly-Gly-Met peptide. They go on to use these antibodies in conjunction with mass spectrometry proteomics to map ubiquitination sites on substrates of UBE2W and show these substrates include UCHL1 and UCHL5. The manuscript seems solid and is well-written, however I have written below a few comments and suggestions for the authors to improve the manuscript.

Antibody engineering

Please include the size / diversity of the libraries obtained and mention the reasoning behind pooling the libraries in 2x2 format as well as the use of scFv format versus Fab. Were there any ScFv positive clones which did not show binding once reformatted to Fab/IgG?

Did the authors test binding of the antibodies to the non-digested purified UCHL1 and UCHL5 proteins produced? If these substrates have disordered N-termini as suggested in the article, the antibodies might be able to recognize these sites on the purified protein.

Please mention which SEC column was used for IgG purification or add a reference.

Structural biology

Based on the pdb validation report, the structure quality is decent but could be improved. I suggest the authors run the structure through the pdb-redo server if they have not already and see if this helps with refining the structure. Some of the clashes with high overlap are with water molecules, and I suggest removing those water molecules unless there is strong density evidence for these molecules.

Please mention which software was used to calculate the buried surface area between the GGM peptide and Fab.

The sequences of Fab 2B12 and 1C7 are quite different, and Fig.2.b. is not clear in showing how these differences might impact the selectivity profile. Fig.2.a and 2.b. are so similar they look redundant. I suggest adding additional panels to Fig.2. showing the electrostatic surfaces of the binding sites for these Fabs (APBS server) and a model of 2B12 binding to GGW based on the 1C7 structure. Playing with depth cue, transparency and surface representation might help clarify Fig. 2.

Similarly, supl.Fig.2.c. might be made clearer by showing the surface area with high transparency.

Other

Please speculate on why the modification of endogenous UCHL5 could not be detected.

Please add the sensorgram data for the BLI experiments to the supplementary.

Typos

“vendor” was not replaced by the vendor name for the anti-rabbit Fc HRP secondary antibody.

Author CS is missing from the author contributions' list.

Reviewer #3 (Remarks to the Author):

In the manuscript by Davies and colleagues, the authors report on the development and characterization of monoclonal antibodies that specifically recognize an N-terminal diglycine motif on peptides that may arise from N-terminal directed ubiquitination. The authors evaluate the specificity of the N-terminal residue using elisa assays and show some broad specificity with some N-terminal residues showing no binding (proline as a notable example). Importantly, the authors demonstrate that these newly generated antibodies do not recognize the isopeptide-linked diglycine modified peptides that the established anti-K-GG antibody binds to. They then go on to determine the nature of this specificity through structural analysis of antibody recognition of the GGM peptide. The authors then employ their antibodies toward identification of endogenous N-terminal ubiquitination events using immunoaffinity enrichment followed by LC-MS/MS-based peptide identification and quantification. Here they identify a small number of putative N-terminal ubiquitination substrates with most being derived from GG-M peptides. To increase abundance of possible N-term ub substrates, the authors overexpressed UBE2W (and RNF4 with not much additional effect) and identified ~100-150 substrates using a variety of quantitative and filtering approaches. They focus on two deubiquitinating enzymes (dubs), UCHL1 and UCHL5, and show that internal lysine-less versions of these dubs can be ubiquitinated in vitro by UBE2W and that UBE2W overexpression results in enhanced mono-Ub modified endogenous UCHL1. Finally, they generate Ub-UCHL1, and Ub-UCHL5 fusion proteins and show that appending ubiquitin to the N-terminus has opposite effects on their dub activity in vitro and upon overexpression (at least of UCHL1 when assaying mono ubiquitin levels in cells).

Overall, the study is well-executed, and the stated results are well-substantiated by the data. They authors have generated a useful set of antibody reagents that can be employed to understand the extent of N-terminal ubiquitination (an often-overlooked component of protein ubiquitination) and what cellular contexts N-terminal ubiquitination might be utilized. They nicely validate the antibody reagents and utilize a commendable cornucopia of assays to benchmark their reagents.

One of more impactful findings is the difficulty the authors encountered in identifying N-terminal ubiquitinated proteins without UBE2W overexpression. It would be useful for the authors to comment on this finding as to whether this is a shortcoming of the reagents (not strong enough affinity for these peptides) or that N-term ubiquitination is just very rare and lowly abundant.

Also, the observation that most of the peptides identified from the IP-MS experiments are derived from GG-M peptides is interesting but may be limiting the ability to interrogate more abundant N-term ubiquitination events as the MetAP activity in cells is quite high. The authors nicely discuss this limitation, but it would be useful for them to expand on this a bit and cite some N-term omics

studies as to how many proteins retain their initiator methionine after translation. If this number is quite low, it would be helpful to report that observation as this would spur development of other N-term ub peptide (GG-P) antibodies to get a broader view of the prevalence of N-term ubiquitination.

There are three areas which could use some additional investigation to more fully develop some of their findings. However, none of these are essential and are only recommendations.

First, the authors comment on the discrepancy in the field whether N-term ubiquitination targets proteins for degradation. They attempt to address this by looking at the stability of mono-ubiquitinated UCHL1 (presumably N-term modified) upon UBE2W overexpression and show no enhanced instability of the mono-ub form versus the unmodified form. This result is intriguing but, as it is only one substrate, cannot be used to argue that N-term ubiquitination does not target proteins for degradation. The authors are well-positioned to perform a larger analysis by looking at how the abundance of N-term modified peptides changes upon proteasome inhibition. I was surprised that the authors didn't perform a proteomic analysis of N-term ubiquitination upon proteasome inhibition as this would directly inform whether these substrates are targeted for degradation.

Second, the authors utilize lysine-less versions (K0) of putative substrates to show that they are UBE2W substrates upon co-overexpression. While I appreciate the practicality of this approach, this does not really inform if these proteins are N-terminally ubiquitinated within cells when internal lysine residues are available. A simple comparison of the lysine containing and lysine-less versions would get to the heart of the matter. Even if the lysine-containing versions are not robustly N-terminally ubiquitinated upon UBE2W expression, it is worth reporting this result.

Third, and similar to the second point, is that the direct fusion of ubiquitin to the N-terminus of UCHL1 and UCHL5 obviously impacts ubiquitin binding and dub activity, but these assays merely report on what is a possible functional outcome of an N-terminally ubiquitinated version of these dubs. Whether N-term ubiquitination significantly impacts the function of these dubs is not addressed by these approaches. To address this, the authors could knock down UBE2W or overexpress UBE2W and assay endogenous UCHL1 and UCHL5 activity in lysates using suicide probes. These experiments would directly address if N-term ubiquitination alters the activity of either of these dubs in cells.

Minor issue:

In Figure 1D, the greyscale is saturated for most of the boxes making interpretation of differences, say between GG-M and GG-F peptides impossible. It might be useful to have that box-plot with two different scales to make it easier to see differences at both ends of the spectrum, if possible.

Reviewer #1 (Remarks to the Author):

1. The authors generated antibodies against GGM peptide that can also interact with a subset of GGX peptides. As GGM peptide is also a signature peptide for linear ubiquitin linkages, these antibodies could also enrich such Ub chains (either generated by LUBAC or nascent Ub chains upon translation of tandem/polyubiquitin chains) – please provide data that show how these antibodies recognize linear linkages (under conditions favoring generation of linear ubiquitin chains) and nascent polyubiquitin chains.

We thank the reviewer for this suggestion. While the conditions used for our cellular experiments were not suitable for generating linear Ub chains, we show that the mAbs can recognize the GG-modified N-terminal Ub peptide that would be liberated from linear Ub chains upon trypsin digestion (Supplementary Fig. 5c).

2. Can authors validate putative UBE2W N-terminal ubiquitin-modified substrates also in UBE2W KO cells? Is any N-terminal ubiquitination observed in that context?

We thank the reviewer for this suggestion. HEK293 cells express very low levels of UBE2W, suggesting that it functions either in a context (i.e. stimulus) dependent manner, within a specific subcellular compartment, or within a specialized cell type. Elucidating where specifically UBE2W functions is beyond the scope of this work, although we hope that by identifying and validating putative substrates through overexpression in HEK293 cells, that this work will shed light on future investigations in this area. Additional commentary is provided below in response to Reviewer #3.

3. Figure 4E. Authors state that identified UBE2W substrates show enrichment of the monoubiquitinated form, with certain proteins being modified by polyubiquitin chains. However, in most of the examples in Figure 4E there is a clear difference between UBE2W-overexpressed cells (WT or W144E) and cells only having low endogenous UBE2W levels. However, there is no striking difference between UBE2W wt and mutant cells, as even mutant cells have often smear above substrate. How can authors explain that? How does C91S mutant of UBE2W behave in such validation?

Biochemical ubiquitination assays with UBE2W^{W144E} has been shown to completely abolish substrate binding towards disordered N-termini (Vital et al, Nat Chem Bio. 2015). In our cell culture overexpression studies we observed that with some substrates the expression of UBE2W^{W144E} resulted in the absence of the mono Ub form but the formation of a smear above the substrate. We hypothesize that the W144E mutation is not fully efficient in cells and some substrates may be polyubiquitinated by the combination of UBE2W^{W144E} and unknown E3 ligases by an N-terminal Ub independent mechanism. The C91S mutation would have been an interesting additional control but we believe that

this could also have resulted in the polyubiquitination of the substrates by the same potential mechanism described above.

4. As authors could not observe modification of endogenous UCHL5 in WB, were they able to find the modification of endogenous UCHL5 by MS approaches?

Yes, N-terminal ubiquitination of endogenous UCHL5 was observed in multiple experiments. Included among these was the experiment performed in WT HEK293 cells without overexpression of UBE2W (pilot MS experiment) (Supplementary Data 1).

5. Does Ile44 mutant of Ub fused to N-terminus of UCHL1 or UCHL5 affect in the same way as wild-type Ub the catalytic activity and Ub-binding properties of UCHL1 and UCHL5? Is UCHL1 able to interact in cis with its N-terminal Ub modification? In other words, does the ability of UCHL1/UCHL5 to bind Ub directly affect catalytic activity of these enzymes?

We thank the reviewer for this suggestion and as a part of the revision we performed an experiment in which we compared the monoubiquitin binding properties and the catalytic activities of UCHL1 and UCHL5 alone, N-terminally linked to wild type ubiquitin, and N-terminally linked to an I44A mutant of ubiquitin. We saw that the I44A ubiquitin that is N-terminally linked to UCHL1 causes ~3-fold increase in binding to monoubiquitin, whereas, there was no effect on UCHL5. Regarding the catalytic activity, there are minimal differences between the I44A ubiquitin and the wild type ubiquitin that is N-terminally linked to either UCHL1 or UCHL5. These data suggest that N-terminal ubiquitination on UCHL1 is competing for monoubiquitin binding in cis, however, abolishing this interaction by using the I44A mutant has minimal effect on its catalytic activity, suggesting that the Ub in this context may block the rearrangement of the catalytic triad after binding to the substrate.

6. Is UCHL5 recruitment to 19S proteasome and binding to Rpn13 affected by N-terminal modification?

We thank the reviewer for this suggestion. It will be interesting to understand all of the ways in which N-terminal Ub can affect UCHL5 function. However, our follow up experiments took us in other directions, leaving this query beyond the scope of our current work.

Minor comments

1. To which Institution do all the authors (except 5: Carsten Schwerdtfeger) belong to? Only department names are provided. I assume Genentech?

We have updated the manuscript to state that Carsten Schwerdtfeger is affiliated to Boston Biochem and that all other authors were employees of Genentech at the time the work was completed.

2. Word(s) is missing in the sentence: “Subsequent work demonstrated that N-terminally ubiquitinated proteins do not accumulate significantly upon proteasome inhibition, suggesting that N-terminal ubiquitination might have additional roles beyond promoting proteasome-mediated degradation¹⁹, such as serving as (word missing) assisting in the folding of nascent polypeptides²⁰”

We have updated the manuscript to read:

“Subsequent work demonstrated that N-terminally ubiquitinated proteins do not accumulate significantly upon proteasome inhibition, suggesting that N-terminal ubiquitination might have additional roles beyond promoting proteasome-mediated degradation¹⁹, such as serving as a chaperone in the folding of nascent polypeptides²⁰”

3. Discussion sentence “To date, only one Ub-conjugating enzyme, UBE2W²¹, and one Ub ligase, the linear Ub chain assembly complex (LUBAC)²², have been described to attach Ub or Ub chains at the N-terminus, respectively. “ is a bit misleading, as it suggests that LUBAC can attach Ub chains to a subset of substrates. As far as reviewer knows, LUBAC has only been shown to attach Ub to another Ub molecule, in head-to-tail mode, through peptide bond between Gly76 residue of one Ub and Met1 residue of another Ub. Please rephrase for clarity.

We have updated the manuscript to read:

“To date, only one Ub-conjugating enzyme, UBE2W²¹, which attaches Ub to the N-terminus of proteins, and one Ub ligase, the linear Ub chain assembly complex (LUBAC)²², which attaches Ub to the N-terminus of another Ub molecule, have been described.”

4. Figure 4E. Labeling for HA has been partially cropped in panels 1 and 3.

We have corrected this figure.

Reviewer #2 (Remarks to the Author):

Antibody engineering

1. Please include the size / diversity of the libraries obtained and mention the reasoning behind pooling the libraries in 2x2 format as well as the use of scFv format versus Fab. Were there any ScFv positive clones which did not show binding once reformatted to Fab/IgG?

We thank the reviewer for this recommendation. The material and methods have been updated to include more detail on the generation of the libraries, diversity, and format. We only picked unique scFv clones and all clones maintained binding upon conversion into IgGs.

2. Did the authors test binding of the antibodies to the non-digested purified UCHL1 and UCHL5 proteins produced? If these substrates have disordered N-termini as suggested in the article, the antibodies might be able to recognize these sites on the purified protein.

The antibodies described in this paper recognize an N-terminal GG modification with a free N-terminal amine for binding. This proteotypic motif is only exposed after trypsin digestion, and hence will not be recognizable on intact Ub-UCHL1/5 proteins. Additional evidence has been included in Supplementary Fig. 5c. We performed an ELISA with peptides corresponding to the N-terminus of UCHL1 and UCHL5 and we see that the mAbs only recognized sequences corresponding to the GG-modified N-terminal sequences and not the endogenous N-termini or RGG-modified N-terminal sequences.

Please mention which SEC column was used for IgG purification or add a reference.

We have updated the manuscript.

Structural biology

3. Based on the pdb validation report, the structure quality is decent but could be improved. I suggest the authors run the structure through the pdb-redo server if they have not already and see if this helps with refining the structure. Some of the clashes with high overlap are with water molecules, and I suggest removing those water molecules unless there is strong density evidence for these molecules.

We thank the reviewer for suggestions to see if the structure quality can be improved. We ran the structure through the pdb-redo server, and at a first glance, the R/Rfree appeared to have improved to 0.1884/0.2292 from the original 0.2080/0.2525, which was surprising as it was quite significant, especially when 78 of the water molecules (~60%) that had reasonable, if not perfect density had been removed from the structure during refinement @PDB-REDO. We noted that the Refinement was performed using Refmac by PDB-REDO.

We then took this refined model from PDB-REDO, and calculated the refinement statistics without any further refinement in Phenix, which we used for refinement for this structure previously and the results were strikingly different. For the refined model from PDB-REDO, the R/Rfree calculated using Phenix was now 0.2395/0.2679, which is quite higher than 0.1884/0.2292 reported by PDB-REDO. We then visually inspected the maps, and while the overall structure was not very different, we observed slight changes in rotamers of some residues.

The major result after changes by PDB-REDO was deletion of 78 water molecules which has somewhat imperfect density, and could have had less than ideal distances from the hydrogen bonding atoms. We then deleted these 78 atoms from our original structure and an additional 2 water molecules by inspection (in addition to a couple of other edits to the structure that needed to be done for close contacts upon close inspection) and calculated final refinement statistics using Phenix, which resulted in an R/Rfree of 0.2110/0.2603 (compared to 0.2080/0.2525 with the 80 water molecules included).

In addition, results from MolProbity analysis (in Phenix) are as follows (Original Structure with 80 water molecules removed with a few mother minor changes):

Ramachandran outliers: 0.47%
Ramachandran favored: 93.66%
Rotamer outliers: 0.82%
C-beta outliers: 0
Clash score: 8.15
Overall score: 1.86

Whereas, the results from MolProbity analysis (in Phenix) for the structure after refinement by PDB-REDO are as follows:

Ramachandran outliers: 0.35%
Ramachandran favored: 95.32%
Rotamer outliers: 6.67%
C-beta outliers: 7
Clash score: 4.65
Overall score: 2.19

In addition, for consistency, we decided to stick with refinement and calculation of the final refinement statistics in Phenix for direct comparisons between multiple structures, and NOT compare the statistics from the Phenix refinement with the statistics from PDB-REDO (Refmac).

Taking into account the all the above factors, we concluded that our original structure, with the removal of 78 water molecules suggested by PDB-REDO and the 2 molecules we identified by inspection of maps, with some edits to remove other close contacts, upon a final refinement run in Phenix, is likely the most optimal structure, given the moderate resolution of the crystal structure @2.85 Angstroms.

The listed geometry validation issues in the validation report are expected (cis-peptides, and the close contacts of sulfur atoms in Cysteine residues that form disulfide bonds). All disulfide bond lengths are in the expected range.

We have now uploaded this updated structure to the PDB (7MFR) and the validation report is included in the manuscript submission. We will release coordinates once the manuscript is accepted.

4. Please mention which software was used to calculate the buried surface area between the GGM peptide and Fab.

We have updated the manuscript.

5. The sequences of Fab 2B12 and 1C7 are quite different, and Fig.2.b. is not clear in showing how these differences might impact the selectivity profile. Fig.2.a and 2.b. are so similar they look redundant. I suggest adding additional panels to Fig.2. showing the electrostatic surfaces of the binding sites for these Fabs (APBS server) and a model of 2B12 binding to GGW based on the 1C7 structure. Playing with depth cue, transparency and surface representation might help clarify Fig. 2.

We thank the reviewer for this suggestion. The intention behind presenting the two panels in figure 2 was to showcase the binding site for the GGM peptide from different viewpoints. However, we agree that the two panels are somewhat redundant due to the small size of the molecule (GGM peptide) – one panel provides sufficient clarity and we have retained only one version of the close up of GGM bound to 1C7 fab.

1C7 and 2B12 do indeed appear significantly different in terms of the sequence. We mapped the differences between the variants onto the 1C7-GGM crystal structure and analyzed the effect of the amino acid substitutions on the structure of 1C7 fab:

(a) A large percentage (~85%) of the residues (21/25 in the Light Chain, LC and 24/27 in the Heavy Chain, HC) are either on the surface of the fab, exposed to solvent, or, significantly further from the GGM peptide and hence are not expected to affect the GGM binding site characteristics. The substitutions to bulkier residues in most cases are complemented by substitutions to compact residues directly next to them, especially in the interior of the protein and elsewhere where residues pack against each other. Out of the residues that differ between 1C7 and 2B12, we identified 4 residues in the LC (A34, Y49, T91, L96) and 3 residues in the HC (Y32, I50, T93) of 1C7, that directly interact with the GGM peptide.

(b) In the LC, substitutions A34S, Y49F interact with the GG-part of the GGM peptide, and the substitutions likely do not have an appreciable effect on the GGM peptide binding. The bulkier T91L substitution is accommodated by the simultaneous compact R32H substitution without affecting the peptide binding. Only the L96N substitution slightly increasing the volume and increasing hydrophilicity of the pocket where the Methionine side-chain binds.

(c) In the HC, substitutions Y33W, I50A interact with the GG-part of the GGM peptide, and the substitutions likely do not have an appreciable effect on the GGM peptide binding based on their orientation. Substitution T93V directly coordinates with the methionine side-chain and increase hydrophobicity of the pocket, without altering the volume significantly.

So, it appears that L96N in LC and T93V in HC result in recognition of W in GW vs M in GGM (with L96 and T93). Of course, beyond the primary coordination shell, given the secondary and ternary shell residue differences, it is possible that other conformational change may occur to accommodate the W vs M, but modeling those changes or solving a structure of 2B12 in complex with GW peptide is beyond the scope of this manuscript.

We also looked at generating electrostatic surfaces using the APBS server / Pymol, but found that it was practically impossible to see any differences, as the L96N in HC and T93V in HC for a model of 2B12, were so deep inside the pocket for the Methionine sidechain in GGM, and the top of the pocket was closed off by residues coordinating the N-terminus and the two glycines. Plus, the electrostatic changes with those two mutations are subtle – there is no charge reversal, for example – in the GGM pocket. So, we did not include a figure showing the same.

To clarify the potential of the slightly altered pocket to enable recognition of GW vs GGM, we show in Figure 2b and 2c, a comparison of the GGM pocket vs GW pocket respectively, without the peptide bound, with the empty pockets, with depth cue and surface representation to highlight the slight differences that could be modeled.

6. Similarly, suppl.Fig.2.c. might be made clearer by showing the surface area with high transparency.

For Supplementary Figure 2c, since it is a model where we are trying to model potential clashes, as opposed to additional space created (see 5 above), we felt it would be clearer if we showed the figure as is, with the short distances as shown, to highlight the incompatibility of the current 1C7 fab structure with GGP peptides.

7. Please speculate on why the modification of endogenous UCHL5 could not be detected.

One challenge with data dependent shotgun sequencing is that individual peptide ions ‘compete’ for the attention of the detector with all other peptides that co-elute and get ionized together. While these GGX enrichment antibodies do a

remarkable job increasing the relative signal of N-terminally ubiquitinated peptides, one challenge that remains is that many proteins code for proteotypic peptides that begin with a comparable GGX sequence and are enriched by our antibodies. In the early experiments, we took advantage of this to define the specificity of our antibodies for this N-terminal sequence motif. While useful for this purpose, the most likely explanation for 'absence of evidence' with respect to the UCHL5 GGX peptides is that they co-elute with internal GGX peptides. This issue was the reason that we pivoted away from using TMT for GGX experiments and back to label free quantitation.

8. Please add the sensorgram data for the BLI experiments to the supplementary.

We have updated the manuscript to include the entire BLI sensorgram data.

9. "vendor" was not replaced by the vendor name for the anti-rabbit Fc HRP secondary antibody.

We have updated the manuscript.

10. Author CS is missing from the author contributions' list.

We have updated the manuscript.

Reviewer #3 (Remarks to the Author):

Comments:

1) One of more impactful findings is the difficulty the authors encountered in identifying N-terminal ubiquitinated proteins without UBE2W overexpression. It would be useful for the authors to comment on this finding as to whether this is a shortcoming of the reagents (not strong enough affinity for these peptides) or that N-term ubiquitination is just very rare and lowly abundant.

We thank the reviewer for this comment and discuss several possibilities in the discussion.

1) Existing data indicates that cell-specific factors may contribute. For example, HEK293 cells have very low UBE2W expression and use of cell types with higher levels would be predicted to yield more potential substrates. Our data suggests that some additional ligase(s) can also mediate N-terminal Ub and identification of these ligase(s) can better identify cell types in which N-terminal Ub plays a role in cell signaling.

- 2) It is possible that N-terminal ubiquitination is a stimulus-dependent phenomenon. In similar experiments performed with Parkin overexpression (Bingol et al. Nature 2014, Cunningham et al. Nat. Cell Biol 2015), very few substrates are identified in the absence of a ligase activating stimulus – in that case, mitochondrial depolarization.
- 3) The availability of free N-termini also likely plays a role. In certain contexts, much larger potential pools of substrates could exist such as modulation of N-terminal acetylation, induction of stress responses or stimulation of intracellular proteolytic signaling events, such as caspase activation.
- 4) It remains possible that N-terminal ubiquitination is occurring within a subcellular compartment within the cell. For example, this may be in proximity to a specific protein complex or subcellular structure. We are hopeful that the identification and validation of putative substrates will reveal the factors controlling the generation of N-terminal ubiquitin modifications.
- 5) We also hypothesize that N-term Ub might be a dynamic modification in which active deubiquitination regulates its abundance. In a previous study (Keusekotten et al. Cell 2013), endogenous linear ubiquitin chains are not visible in the whole cell lysate unless OTULIN is mutated to be catalytically inactive. Furthermore, linear chains are more visible when endogenous OTULIN is knocked down. Therefore, unless we block or downregulate deubiquitinases it is going to be very difficult to see a massive accumulation of N-term Ub substrates.

2) Also, the observation that most of the peptides identified from the IP-MS experiments are derived from GG-M peptides is interesting but may be limiting the ability to interrogate more abundant N-term ubiquitination events as the MetAP activity in cells is quite high. The authors nicely discuss this limitation, but it would be useful for them to expand on this a bit and cite some N-term omics studies as to how many proteins retain their initiator methionine after translation. If this number is quite low, it would be helpful to report that observation as this would spur development of other N-term Ub peptide (GG-P) antibodies to get a broader view of the prevalence of N-term ubiquitination.

We thank the reviewer for this suggestion. The literature does suggest that a high number of initiator Met can be removed, and thus, expanding of the antibody toolset would have value to further mine these proteins for potential N-term Ub events. We have added the sentence below to the discussion.

“Proteomics studies also indicate that up to ~60-70% of initiator Met can be removed, suggesting expanding our antibody toolset to other GGX peptides may identify additional N-terminal ubiquitination sites (Yeom et al. Sci Rep 2017; Lange et al. J Proteome Res 2014).”

There are three areas which could use some additional investigation to more fully develop some of their findings. However, none of these are essential and are only

recommendations.

3) First, the authors comment on the discrepancy in the field whether N-term ubiquitination targets proteins for degradation. They attempt to address this by looking at the stability of mono-ubiquitinated UCHL1 (presumably N-term modified) upon UBE2W overexpression and show no enhanced instability of the mono-ub form versus the unmodified form. This result is intriguing but, as it is only one substrate, cannot be used to argue that N-term ubiquitination does not target proteins for degradation. The authors are well-positioned to perform a larger analysis by looking at how the abundance of N-term modified peptides changes upon proteasome inhibition. I was surprised that the authors didn't perform a proteomic analysis of N-term ubiquitination upon proteasome inhibition as this would directly inform whether these substrates are targeted for degradation.

We thank the reviewer for this thoughtful suggestion and as part of the revision performed an experiment looking at four cellular conditions: UBE2W OE +/- Btz. This experiment was performed using label free quantification to evaluate whether proteasome inhibition impacted the abundance of N-terminally modified "GGX" peptides that were identified and quantified. Remarkably, only 12 out of the 236 substrates identified in this experiment displayed an increase in abundance upon Btz treatment. It is important to note, that UBE2W substrates targeted to the proteasome by other enzymes of the ubiquitin system, but not UBE2W, would be expected to display this quantitative pattern. These data provide evidence that N-terminal ubiquitination functions autonomously from proteasomal degradation for most, if not all substrates, and suggests yet another non-canonical role for the ubiquitin system in controlling biochemical processes within the cell.

4) Second, the authors utilize lysine-less versions (K0) of putative substrates to show that they are UBE2W substrates upon co-overexpression. While I appreciate the practicality of this approach, this does not really inform if these proteins are N-terminally ubiquitylated within cells when internal lysine residues are available. A simple comparison of the lysine containing and lysine-less versions would get to the heart of the matter. Even if the lysine-containing versions are not robustly N-terminally ubiquitinated upon UBE2W expression, it is worth reporting this result.

We thank the reviewer for this suggestion. We repeated this experiment with lysine-containing versions of the putative substrates and we were unable to observe clear bands corresponding to the N-terminally monoUb species. We hypothesize this is due to the addition of Ub to lysines via other ligases, which results in smearing the signal observed with the K0 version across the gel. Please see the following pasted figure.

5) Third, and similar to the second point, is that the direct fusion of ubiquitin to the N-terminus of UCHL1 and UCHL5 obviously impacts ubiquitin binding and dub activity, but these assays merely report on what is a possible functional outcome of an N-terminally ubiquitinated version of these dubs. Whether N-term ubiquitination significantly impacts the function of these dubs is not addressed by these approaches. To address this, the authors could knock down UBE2W or overexpress UBE2W and assay endogenous UCHL1 and UCHL5 activity in lysates using suicide probes. These experiments would directly address if N-term ubiquitination alters the activity of either of these dubs in cells.

We thank the reviewer for this thoughtful suggestion and as a part of the revision we performed an experiment in which we used the Ubiquitin-Vinyl Sulfone (Ub-VS) suicide probe to assay UCHL1 activity in lysates with and without UBE2W overexpression. Without UBE2W expression, we saw that endogenous UCHL1 rapidly reacts with the Ub-VS. Similar results were observed when we overexpressed UBE2W, however, the N-term Ub species of UCHL1 at time 0 (product of UBE2W overexpression) was not able to react with the Ub-VS probe. This was evident by the lack of accumulation of the UCHL1 bound to two ubiquitin moieties. Consistent with our experiments with purified proteins, these results suggest that only the free endogenous UCHL1 is reacting with the Ub-VS, and not, the ubiquitinated UCHL1. These data have been added in Supplementary Fig. 6c.

Minor issue:

6) In Figure 1D, the greyscale is saturated for most of the boxes making interpretation of differences, say between GG-M and GG-F peptides impossible. It might be useful to have that box-plot with two different scales to make it easier to see differences at both ends of the spectrum, if possible.

We have modified Fig. 1D to have a double gradient help decipher the subtle differences between peptides. In many cases the ELISA signal was saturated and thus cannot distinguish differences between the strong binding peptides.

REVIEWERS' COMMENTS

Reviewer #1 (Remarks to the Author):

The authors have addressed most of my concerns. I would like to thank them for performing additional experiments to improve their manuscript.

1. The authors provide evidence that their antibody indeed can recognize linear ubiquitin linkages.
2. As HEK293 cells express low levels of UBE2W, I agree that performing experiment in the context of UBE2W KO is not necessary at this point.
3. Figure 4E. The authors do not really provide any experimental explanation why W144E mutant of UBE2W leads to polyubiquitin smear above UBE2W substrates. They also decline to perform experiment with C91S mutant claiming that "this could also have resulted in the polyubiquitination of the substrates by the same potential mechanism described above." I do not find it satisfactory and I think that the authors should at least discuss this in the Discussion section, since leaving such data in Figure 4E without providing even a hypothetical explanation is not optimal.
4. Concern addressed.
5. Concern addressed.
6. Concern addressed.

All the minor comments were addressed as well.

Reviewer #2 (Remarks to the Author):

I appreciate the extra work spent on structure quality and analysis and I am satisfied with the modifications brought to the manuscript as well as the authors' responses.

Nadia Leloup

Reviewer #3 (Remarks to the Author):

The authors have successfully addressed all previous concerns.

Review #1:

3. Figure 4E. The authors do not really provide any experimental explanation why W144E mutant of UBE2W leads to polyubiquitin smear above UBE2W substrates. They also decline to perform experiment with C91S mutant claiming that “this could also have resulted in the polyubiquitination of the substrates by the same potential mechanism described above.” I do not find it satisfactory and I think that the authors should at least discuss this in the Discussion section, since leaving such data in Figure 4E without providing even a hypothetical explanation is not optimal.

We now elaborate on this point by including the following paragraph in the discussion:

UBE2W strictly monoubiquitinates protein substrates at their N-termini^{21,23,24}, and these modifications can be further modified by the combine function of E2/E3 complexes into N-terminally linked polyubiquitin chains²³. Our *in vitro* ubiquitination assays confirmed that UBE2W is only able to monoubiquitinate substrates (Fig. 5c). However, overexpression studies in cells demonstrated that UBE2W's substrates can be both mono- and polyubiquitinated in cells, confirming previous literature that monoubiquitin on the N-terminus can be further modified to generate polyubiquitinated species (Figure 4C). Interestingly, expression of *UBE2W*^{W144E}, a mutant that lacks the ability to bind to substrates²⁴, resulted in the absence of the monoubiquitinated forms, but largely maintained the polyubiquitinated species (Fig. 4C). We hypothesize that some substrates may be N-terminally ubiquitinated by UBE2W and elaborated into polyubiquitin chains by other E2 and ubiquitin ligase pairs. Future cellular analysis utilizing UBE2W and catalytic dead mutants of UBE2W and other E2s may provide a better mechanistic view into N-terminally linked polyubiquitin chains and the identification of ubiquitination machinery that cooperates with UBE2W.